# Development and Assessment of High-Resolution Radar-Based Precipitation Intensity-Duration-Curve (IDF) Curves for the State of Texas

**Dawit T. Ghebreyesus *** and **Hatim O. Sharif**

Department of Civil and Environmental Engineering, University of Texas at San Antonio, 1 UTSA Circle, San Antonio, TX 78249, USA; hatim.sharif@utsa.edu
* Correspondence: dawit.ghebreyesus@my.utsa.edu

**Abstract:** Conventionally, in situ rainfall data are used to develop Intensity Duration Frequency (IDF) curves, which are one of the most effective tools for modeling the probability of the occurrence of extreme storm events at different timescales. The rapid recent technological advancements in precipitation sensing, and the finer spatio-temporal resolution of data have made the application of remotely sensed precipitation products more dominant in the field of hydrology. Some recent studies have discussed the potential of remote sensing products for developing IDF curves. This study employs a 19-year NEXRAD Stage-IV high-resolution radar data (2002–2020) to develop IDF curves over the entire state of Texas at a fine spatial resolution. The Annual Maximum Series (AMS) were fitted to four widely used theoretical Extreme Value statistical distributions. Gumble distribution, a unique scenario of the Generalized Extreme Values (GEV) family, was found to be the best model for more than 70% of the state's area for all storm durations. Validation of the developed IDFs against the operational Atlas 14 IDF values shows a ±27% difference in over 95% of the state for all storm durations. The median of the difference stays between −10% and +10% for all storm durations and for all return periods in the range of (2–100) years. The mean difference ranges from −5% for the 100-year return period to 8% for the 10-year return period for the 24-h storm. Generally, the western and northern regions of the state show an overestimation, while the southern and southcentral regions show an underestimation of the published values.

**Keywords:** IDF; NEXRAD Stage-IV; Texas; precipitation frequency

## 1. Introduction

The non-stationarity nature of climatic variables requires constant updating of engineering design parameters, especially those that are sensitive to weather and climate extremes. Currently, this issue has become more clear than ever with the increased frequency of extreme weather events such as floods and droughts as a result of the changing climate [1]. The increased frequency of extreme flooding is caused by higher evaporation due to increased atmospheric temperature and the heightened capacity of the atmosphere to hold moisture [2,3]. The conterminous United States experienced a rapid rise in the number of extreme one-day rainfall events with 9 out of the top 10 years of record-breaking one-day events occurring after 1990 [4]. Moreover, the cyclone intensity has dramatically increased in the last two decades, with cyclones responsible for the 6 of the most active 10 years since 1950 happening in the last 20 years, according to the Accumulated Cyclone Energy (ACE) Index [5]. The aforementioned anomalies require the adjustment of the current design parameters with up-to-date rainfall datasets with adequate spatio-temporal resolutions.

The Intensity–Duration–Frequency (IDF) curve is a graphical representation of the probability of the occurrence of extreme precipitation developed based on extreme probability functions. IDF curves relate rainfall intensity, frequency, and duration, all presented in a set of curves. These curves are widely used to obtain the intensity (or depth) of

extreme precipitation to force hydrologic models and then estimate the peak discharge and runoff volume that the structures are expected to withstand during their lifetime [1]. The conventional IDF development process uses historical observations of annual extreme precipitation from rain gauges fitted to the theoretical extreme value distributions [6]. The procedure significantly depends on the length and quality of the historical precipitation record. Examples of recent international studies that have used rain gauge data to construct IDF curves include De Paola [7] in three African countries, Al-Amri [8] in Saudi Arabia, Noor [1] in Malaysia, and Sherif [9] in the United Arab Emirates. The main limitation of such studies is the sparseness of the rain gauge network used, which leaves large portions of the ungauged area to rely on interpolation and extrapolation [10]. In addition, the length of the historical records often differs significantly from one rain gauge to another. To address this limitation, some studies were conducted to transform the point IDF values from the rain gauges to attain spatial IDF variability. For example, the transformation based on the spatial correlation structure of rainfall [11], radar-based areal reduction factors (ARFs) as used in Overeem [12], and a flood frequency analysis framework based on stochastic storm transposition by Wright [13]. IDF development has a strong statistical foundation, and IDF estimates are widely accepted by engineers and hydrologists. However, the lack of long historical records and sparsely distributed in situ observations makes the development of IDF challenging, especially in undeveloped parts of the world [14], prompting engineers to use some interpolation and adjustment techniques such as area reduction factors.

Recently, with the advancement of satellite sensors, high-resolution spatio-temporal precipitation products are becoming globally available. The Global Precipitation Measurement (GPM) products at ~10km and 30 min resolution, the Climate Prediction Center morphing technique (CMORPH) at 8km and 30 min, and the Remotely Sensed Information Using Artificial Neural Networks (PERSIANN) at 0.25 degrees and 1 h are some of the widely used global satellite-based precipitation products. These products provide important details about the spatio-temporal variability of precipitation, which is a very intermittent variable. Moreover, the gridded format of these precipitation products reduces the need for areal interpolation. Furthermore, since most of the sensors used in radar-based applications are identical and because satellite-based applications use a single sensor, the heterogeneity that is introduced by using different types of rain gauges is reduced. Recent studies conducted using remote-sensing products include Ombadi [14], who used the Precipitation Estimation from PERSIANN—Climate Data Record (PERSIANN-CDR); Courty [15], who used hybrid deterministic reanalysis data; Sun [16], who used Global Satellite Mapping of Precipitation (GSMaP) data; and Marra [6], who used CMORPH estimates over the eastern Mediterranean. The main limitation of analyses based on remotely sensed precipitation products is their limited record length. These datasets have only become available recently and only cover the last two to three decades at most. Moreover, their accuracy is also another issue, with many validation studies noting their failure to capture heavy storm events. These limitations also apply to NEXRAD radar-based precipitation data. However, some recent studies have shown that their performance is improving due to the constant upgrading of the algorithms and technological advancement of the sensors [14,17]. Additionally, the NEXRAD Stage-IV data were found to be more accurate than satellite-based precipitation products and were used as reference data in many studies [18–20].

The frequency and magnitude of extreme events caused by rising temperatures and heavy rainfall such as floods and drought are on the rise and are expected to continue to increase as a result of climate change [21]. The change in the frequency and intensity of extreme precipitation in a region requires an immediate updating of IDF curves. Rodríguez [22] demonstrated that daily rainfall is expected to rise significantly in Barcelona, Spain, especially for storms with a return period longer than 20 years. Another study conducted using the projected extreme rainfalls from the nine global climate models (GCMs) suggests the need to modify IDFs for all return periods with higher intensities in Thailand [23]. These studies suggest that IDFs are dynamic and are evolving rapidly due

to climate change. Moreover, engineers are strongly advised to employ the latest IDFs, especially when designing structures with a long lifetime. The main issue of employing rain gauges to develop IDF is the bias that is introduced by the uneven distribution of the rain gauges, as most of them are located in densely populated areas [24]. These aforementioned reasons and the recent advancements in geospatial processing techniques make the potential of remote-sensing-based IDF development very interesting.

The main objective of the present study is to use the high-resolution US National Weather Service (NWS) Next Generation Weather Radar (NEXRAD) Stage-IV precipitation data to develop IDF curves with a high spatial resolution. In addition to the higher resolution, the method followed in this study will simplify the process of developing and updating the current IDFs as frequently as possible to keep up with the changing climate. Furthermore, the study is particularly interesting because it is conducted in the state of Texas, a state that has experienced a significant increase in the magnitude and frequency of extreme weather events, including hurricanes and major floods, in recent years. The timeframe of the study covers 19 years ranging from 2002 to 2020. The final IDFs obtained from this research were compared to the latest version of the IDFs developed by the National Oceanic and Atmospheric Administration (NOAA) for the state of Texas. The study was conducted to assess the feasibility of employing high-resolution Stage-IV data in developing high-resolution IDF curves. The main limitation of the study is the short length of the NEXRAD Stage-IV dataset.

The manuscript is arranged as follows: in Section 2, a description of the study area and the datasets used is provided. In Section 3, the methodology is presented followed by all of the necessary equations and assumptions. The results of the study and a comparison of the developed IDFs by NOAA to the latest standards will be discussed in Section 4. Finally, the summary of the study and the main conclusions are provided in Section 5.

## 2. Study Area and Dataset

### 2.1. Study Area

Texas is located in the southeastern part of the U.S., bounded from 93°31′W to 106°38′W longitude and from 25°50′N to 36°30′N in latitude. Texas is the second-largest state in the U.S. and the largest in the conterminous U.S. The climate of the state varies from warm and temperate in the east to an arid desert climate in the west. Figure 1 shows the location of the study area and spatial distribution of precipitation over Texas. The proximity of Texas to the Gulf of Mexico makes it prone to extreme storm events that are mostly caused by tropical storms and hurricanes. About one-quarter of the hurricanes (1851–2004) that hit the mainland U.S. originating from the Atlantic Basin made their landfall in the state of Texas. In the last couple of decades, the state has witnessed an increase in annual precipitation, especially in the eastern parts of the state. The area receiving annual precipitation higher than 1500 mm increased from 7000 $km^2$ to about 51,000 $km^2$, a more than seven-fold increase [25]. On the other hand, the coastal counties of the state have experienced a surge in population over the last half-century, an increase of more than 150%, which is ranked the 4th highest in the US according to the Census Bureau. The analysis also examines precipitation characteristics over six of the major metropolitan areas of Texas, namely, Dallas-Fort Worth (1484 pixels), Houston (1006 pixels), San Antonio (1095 pixels), McAllen (226 pixels), El Paso (839 pixels), and Amarillo (814 pixels), shown in Figure 1. Coastal Texas is also a key player in the economy of the country, as the Gulf of Mexico is one of the most important regions in the US for energy infrastructure, hosting 51% of the US natural gas processing plant capacity and 45% of the total US refining capacity on its coast [26]. Federal offshore oil production in the Gulf of Mexico also accounts for 17% of total U.S. crude oil production.

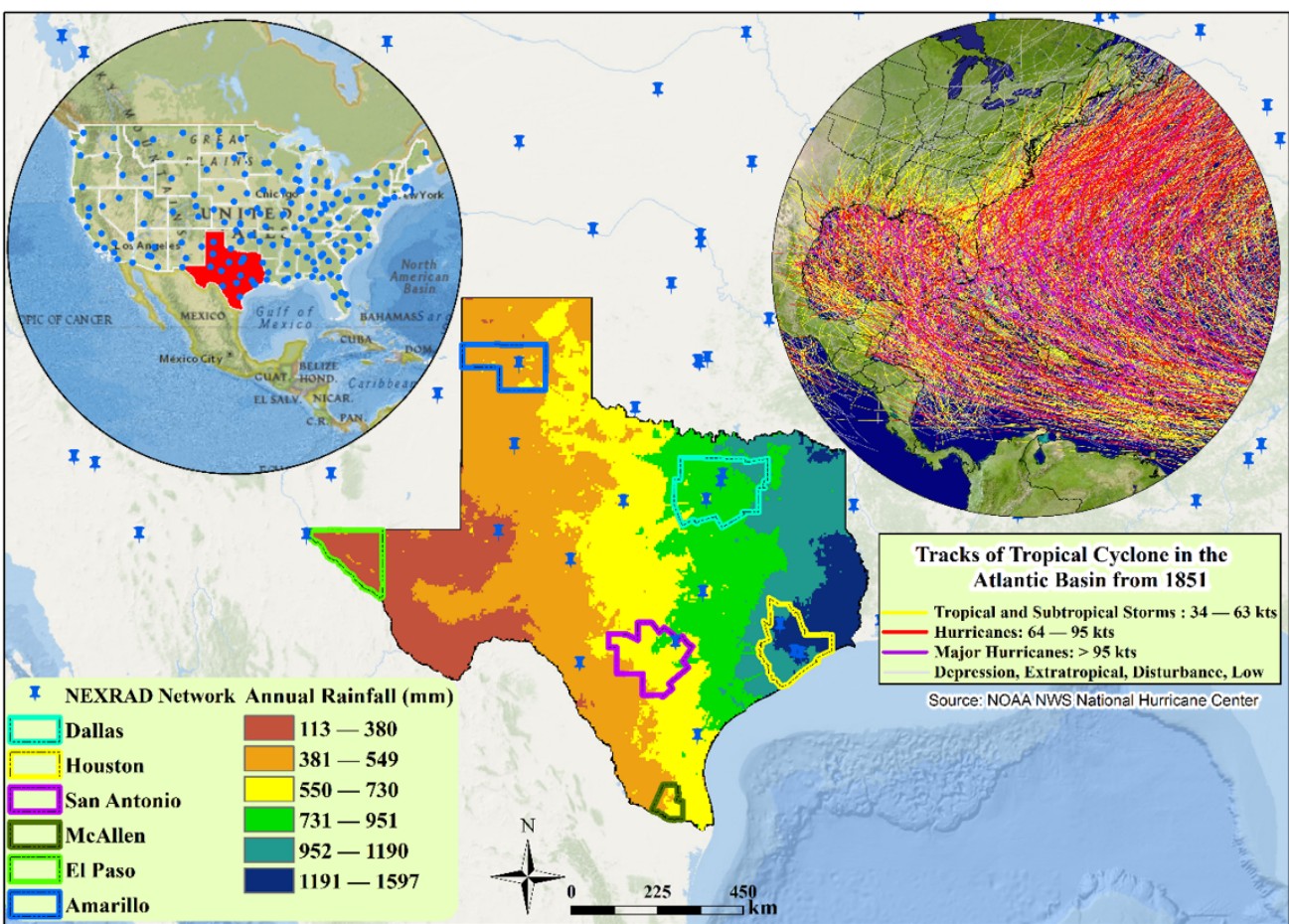

**Figure 1.** Annual average rainfall over the study area based on NEXRAD Stage-IV data (2002–2020) and the trajectory of the tropical cyclones in the Atlantic Basin from 1981 to 2017. The blue points display the spatial distribution of the NEXRAD weather radar network (old, new, and decommissioned). The geographical locations of major metropolitan areas of Texas are also shown.

### 2.2. Dataset

The NEXRAD Stage-IV precipitation data were used in this research. The NEXRAD Stage-III data were prepared by the 12 River Forecast Centers of the NWS. The United States National Centers for Environmental Prediction (NCEP) developed the Stage-IV product based on the NEXRAD Precipitation Processing Systems by merging the Stage-III products from multiple radars [27]. The spatial distribution of the NEXRAD radar network (184 stations) that includes new, old, and decommissioned stations across the conterminous US is shown in Figure 1. The product is used for many purposes, especially for forcing hydrological models over the conterminous US. The product is processed and aggregated from the native 5–6 min temporal resolution to a gridded national product with a temporal resolution of 1-hr. The product covers the entire conterminous US with a very high spatial resolution of approximately 4 km × 4 km with more than 40,000 grid points in the state of Texas. The temporal coverage of the Stage-IV data spans from 1 January 2002 to present day, and 19 years' worth of data (1 January 2002–31 December 2020) are employed in this study. The data is available to the public from the National Center for Atmospheric Research (NCAR) FTP servers on request. More detailed information about the NEXRAD Stage-IV data can be obtained from their website (https://www.emc.ncep.noaa.gov/mmb/ylin/pcpanl/stage4/ [Access date: 15 June 2021]).

## 3. Methodology

The first and foremost step in any kind of research is to analyze the raw data. Here, the spatio-temporal variability of the NEXRAD Stage-IV data was analyzed to capture the variability of the extreme rainfall events across the state. The temporal evolution of the extreme events was analyzed using the Mann–Kendall trend test to evaluate the significance of the trend in the six major metropolitan areas, namely, Dallas-Fort Worth, Houston, San Antonio, McAllen, El Paso, and Amarillo. The main reason for the selection of these six metropolitan areas is that the majority of the state's population lives in these areas, and their spatial distribution represents the entire state.

The intensity–duration–frequency (IDF) relationship is a well-established method that relates rainfall intensity with its duration and annual frequency. The development of IDF curves involves finding the best statistical distribution that explains the variability of the extreme precipitation values. The process starts with accumulating the precipitation for the desired duration for each year. In this study, 11 storm event durations ranging from 1-h to 2-days were used for the development of the IDFs as the radar product does not provide information for sub-hourly durations. The next step is to obtain the annual maximum series for each storm hour duration and fit the series to a well-established theoretical distribution of extreme values. The most common extreme value distributions were then tested, and the one with the best fit according to a model selection criterion was selected.

The Generalized Extreme Value (GEV) distribution is one of the most common and highly cited probability distributions developed within extreme value theory. GEV distribution is suitable for extreme event analysis in a wide range of applications such as annual floods, rainfall, wind speeds, wave heights, snow depths, and other maxima. The distribution has three parameters known as the location, scale, and shape parameters. Gumbel distribution is another theoretical extreme value probability distribution. Gumbel is also referred to as Generalized Extreme Value Distribution Type-I because it is a special case of GEV used when the shape parameter is equal to zero. The equations of the probability distribution function for GEV and Gumbel, respectively, are provided below.

$$f_{(\mu,\sigma,\xi)}(x) = \exp\left(-\left[1 + \xi\frac{x-\mu}{\sigma}\right]^{\frac{-1}{\xi}}\right) \tag{1}$$

$$f_{(\mu,\sigma)}(x) = \exp\left(-\exp\left(-\frac{x-\mu}{\sigma}\right)\right) \tag{2}$$

where $f(x)$ is the probability distribution function, $\mu$ is the location parameter, $\sigma$ is the scale parameter, and $\xi$ is the shape parameter of the continuous distributions.

The Generalized Pareto (GP) distribution is also another family of a continuous theoretical distribution used for extreme values that are based on three parameters similar to GEV. GP distribution was employed in the modelling of extreme events such as in the modeling annual flood values, precipitation data analysis, and in the analysis of flood frequency. The last Extreme Value probability distribution tested is Exponential Distribution. Exponential Distribution is a special type of the Generalized Pareto (GP) distribution, where both the shape and scale parameters are equal to zero. The equations of the probability distribution function for the GP and Exponential distributions, respectively, are provided below.

$$f_{(\mu,\sigma,\xi)}(x) = \frac{1}{\sigma}\left(1 + \xi\frac{x-\mu}{\sigma}\right)^{(-\frac{1}{\xi}-1)}; \begin{cases} x \geq \mu & for\ \xi \geq 0 \\ \leq x \leq \mu - \frac{\sigma}{\xi} & for\ \xi < 0 \end{cases} \tag{3}$$

$$f_{(\sigma)}(x) = \begin{cases} \frac{1}{\sigma}\exp\left(\frac{-x}{\sigma}\right) & for\ x \geq 0 \\ 0 & for\ x < 0 \end{cases} \tag{4}$$

where $f(x)$ is the probability distribution function, $\mu$ is the location parameter, $\sigma$ is the scale parameter, and $\xi$ is the shape parameter of the continuous distributions.

The performance of all the four distributions was assessed to find the best possible distribution for each pixel. The best fit of the four distributions for each radar pixel was evaluated by using the Akaike Information Criterion (AIC), which was developed by Akaike [28], and the Bayesian Information Criterion (BIC), which developed by Schwarz [29]. This step was applied to all the storm event durations over the entire state. Both AIC and BIC have solid theoretical foundations and are not only widely used as model selection criteria in the field of extreme distributions but are also used for all types of statistical models. The best model is the one with the lowest AIC or BIC values, which are calculated as shown in Equations (5) and (6).

$$\text{AIC} = -2 * \log\left(\hat{L}\right) + 2K \tag{5}$$

$$\text{BIC} = -2 * \log\left(\hat{L}\right) + \log(N) * K \tag{6}$$

where $\hat{L}$ is the maximum likelihood, $K$ the number of parameters to be estimated in the model, and $N$ is the sample size.

After the suitable distribution with the best goodness of fit is selected for each pixel, the respective parameters are estimated using the Generalized Maximum Likelihood Estimation method. This method is the most highly recommended because most of the time, the extreme value analysis is conducted using a small sample size, as in this case and for small sample sizes in hydrologic data. The Generalized Maximum Likelihood was found to provide better results [30,31].

Once the parameters are estimated, a 5-pixel by 5-pixel moving window filter was then applied to reduce the noise in the parameters. This filter was used to minimize the impact of the abnormal backscatter that is not related to precipitation. Moreover, parameter values that are estimated to be outliers (four times outside the inter-quartile range), were eliminated from the window. The Kolmogorov–Smirnov test was then used to test the significance of the fit of the theoretical model [32]. Kolmogorov–Smirnov static quantifies the maximum distance between the cumulative distribution of the theoretical model and the empirical distribution of the raw data. The null hypothesis ($H_o$) of the statistical test states that the two datasets, the Annual Maximum Series (AMS) and model fit values, are from the same continuous distribution while the alternative hypothesis ($H_a$) states that these two datasets are from different distributions. Finally, the IDF curves were developed for the desired return periods using the optimal parameters. The full path flow of the methodology that was followed in this study is illustrated in Figure 2. The entire analysis was done pixel-wise over the whole state of Texas, which includes more than 40,000 pixels of approximately 4 km × 4 km.

The final IDFs were then compared with the IDF maps developed by the NOAA Atlas 14, produced in 2018 [24] for the state of Texas. The NOAA Atlas 14 was published in several volumes and describe estimates of the extreme precipitation frequency for the United States and U.S. affiliated territories. Volume 11 covers the state of Texas. The NEXRAD Stage-IV grid points were compared to the nearest point of the NOAA Atlas 14 product. Atlas 14 estimates are based on data from a network of more than 2000 rain gauges distributed across the state, with record lengths ranging from 12 to 150 years. The NEXRAD Stage-IV dataset used for the study ranges from 2002 to 2020, which provides an overlapping timeframe with most of the NOAA rain gauges from 2002 to 2017.

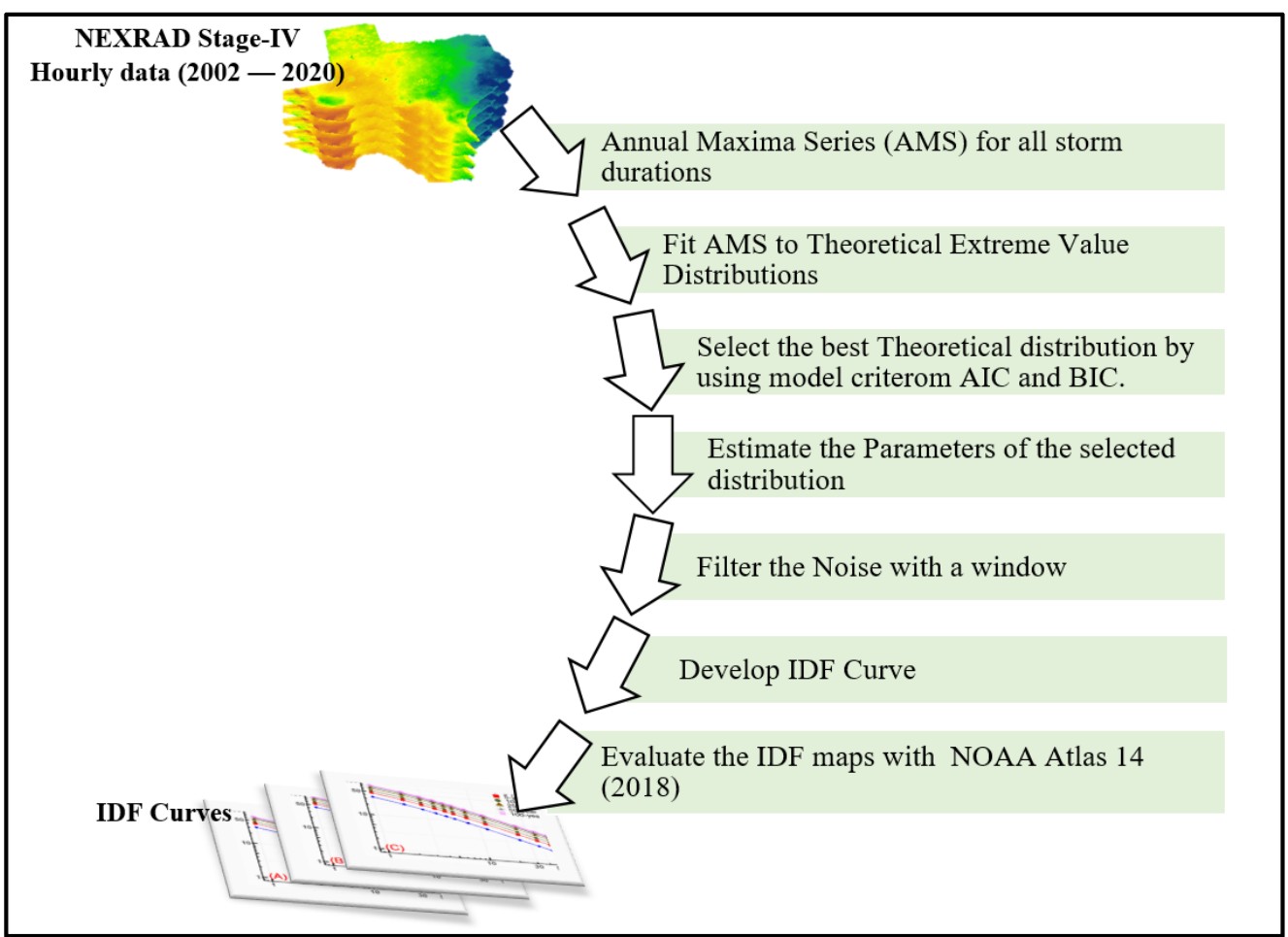

**Figure 2.** The path flow of the methodology followed in developing the IDF curves using the NEXRAD Stage-IV hourly data for the state of Texas.

## 4. Results and Discussion

### 4.1. Spatio-Temporal Variability of the Annual Precipitation Maxima

The annual maxima of the different storm durations based on the Stage-IV radar data (2002–2020) show similar patterns of spatial distribution, as shown in Figure 3. The upper Gulf Coast (around Houston metropolitan area) receives the highest annual maxima, and the lowest values are recorded on the western edge of the state (close to the El-Paso Metropolitan area). However, the spatial variability increases with the storm duration. The average annual maxima for a 1-h storm duration are about 30 mm for the first and third quartiles of 23 mm and 37 mm, respectively. The spatial variability rapidly increases up to the 12-h storm duration and does not change much for the longer durations, as shown in Figure 3C,D. This can be attributed to the infrequency of long-duration storm events (longer than 12 h). This is also shown as a higher rate of increase in the normalized interquartile range (IQR) for the small to medium (from 1-h to 12-h) storm durations with relatively constant normalized IQR for longer durations (1-day, 2-day, and 3-day).

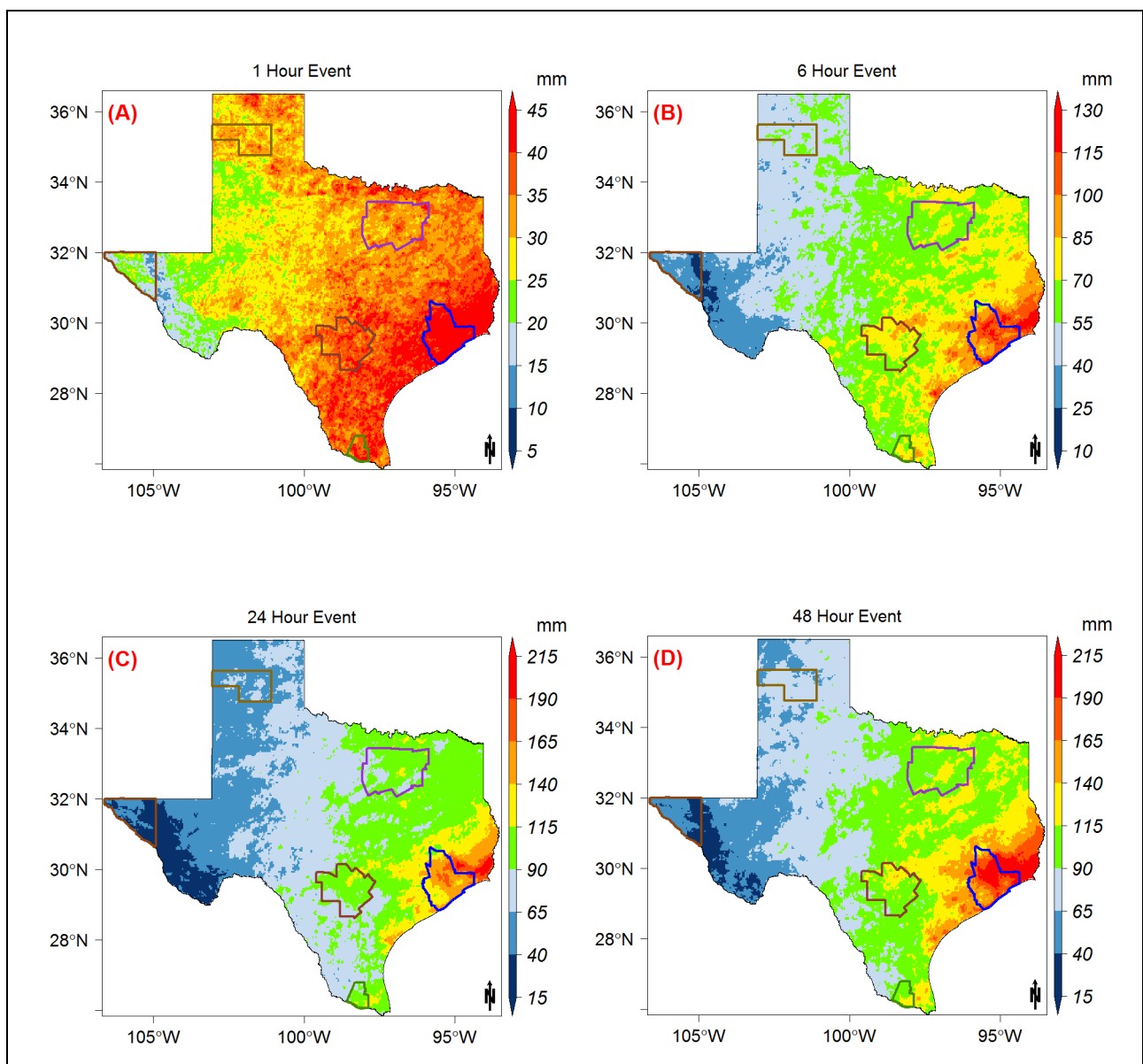

**Figure 3.** Spatial distribution of the average of the annual maxima series for or the (**A**) 1-h, (**B**) 6-h, (**C**) 24-h, and (**D**) 48-h duration storms.

The temporal distribution of the annual maxima series shows that four of the metropolitan areas experienced an increase in the annual maxima in the last two decades (Figure 4). The rate of increase was higher in the metropolitan areas closer to the Gulf of Mexico (Houston and McAllen). However, the trend in Houston was found to be significant according to the Mann–Kendall trend test. San Antonio and Amarillo, which are located in the southcentral and northern parts of the state, respectively, show a decreasing trend of the annual maxima series over the last two decades (Figure 4C,F). The observed increase in tropical cyclone activity in the Gulf of Mexico in the last three decades [4] probably explains the increase in the magnitude and frequency of extreme events in the areas closer to the Gulf of Mexico. El Paso, a metropolitan area located in west Texas that is the driest region of the state, experienced a relatively stable temporal distribution of AMS, except for in 2013 (Figure 4E). Additionally, El Paso was found to be the only metropolitan area to have a peak in 2013. This is related to the extreme precipitation that occurred due to the strong southwesterly monsoon flow over New Mexico and Colorado and two tropical storms in

September 2013 [33]. The event affected parts of New Mexico, southeastern Colorado, and far west Texas.

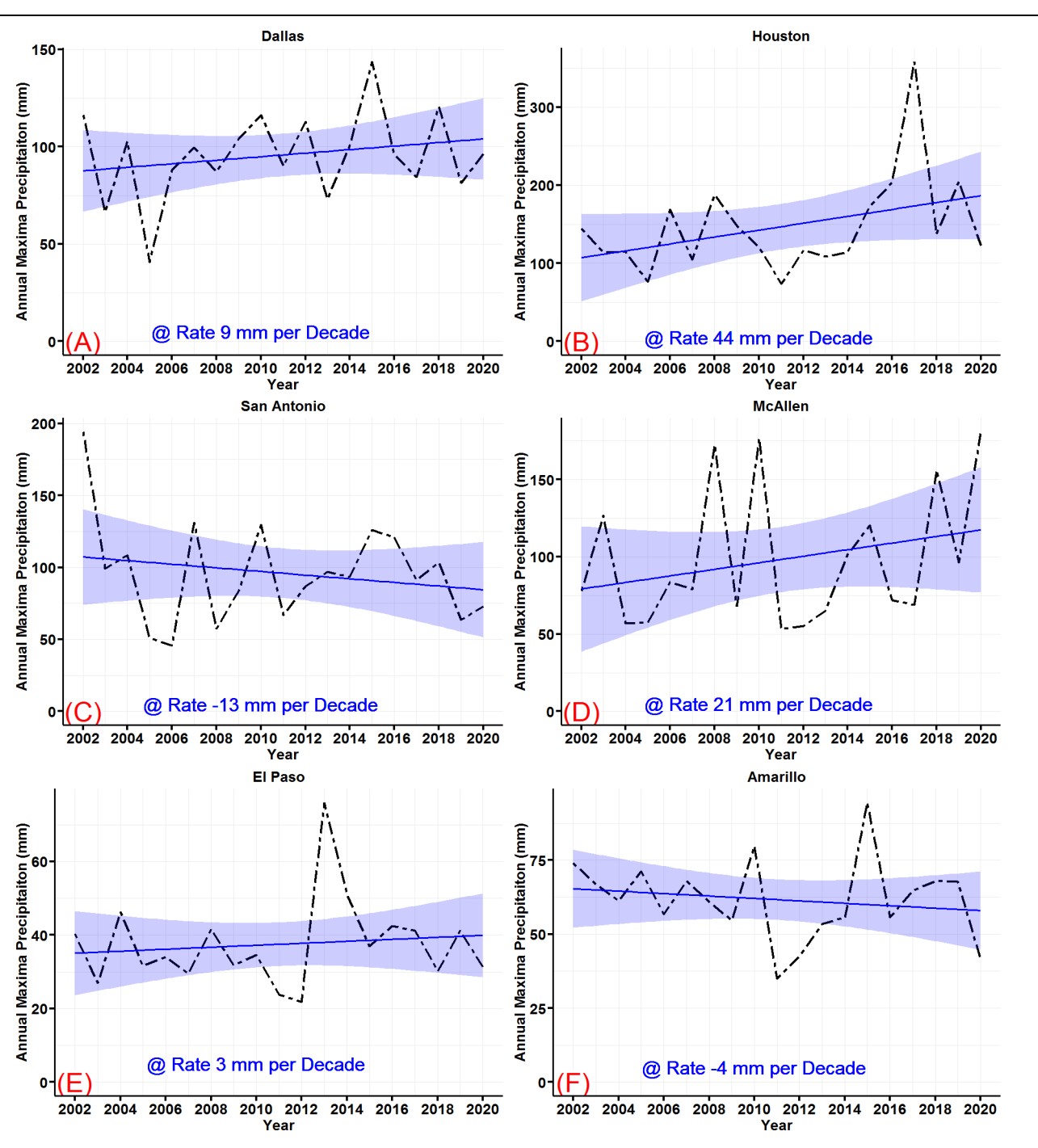

**Figure 4.** Temporal distribution of the 24-h duration storm with trendlines (blue) across the main metropolitan areas of (**A**) Dallas-Fortworth, (**B**) Houston, (**C**) San Antonio, (**D**) McAllen, (**E**) El Paso, and (**F**) Amarillo (shading represents the 95% confidence interval of the trendline using t-distribution).

### 4.2. Statistical Distributions

Traditionally, a single distribution is used across the entire state to reduce the discontinuities in frequency estimation caused by multiple distributions in the spatial and temporal domain [24]. The selection of the best distribution is conducted based on the outcome of the AIC and BIC. In this analysis, violin plots of the AIC and BIC were constructed to

visualize their distributions across all of the durations as shown in Figure 5. The three lines inside the violin represent the three quartiles (25%, 50%, and 75%) of the distribution. When using AIC and BIC for model selection, the value of the parameter does not have significant meaning, however, the model with the lowest values is the model with the best fit. As shown in Figure 5, the GEV and Gumbel distributions have the lowest AIC and BIC medians. Comparing the means and medians of the distributions, the Gumble distribution shows the lowest AIC and BIC for all storm durations. The Exponential distribution has the weakest fit, while the GP distribution shows widely spread values of AIC and BIC. This suggests that a very small area in the state strongly favors the GP distribution, with a poor fit over the majority of the state.

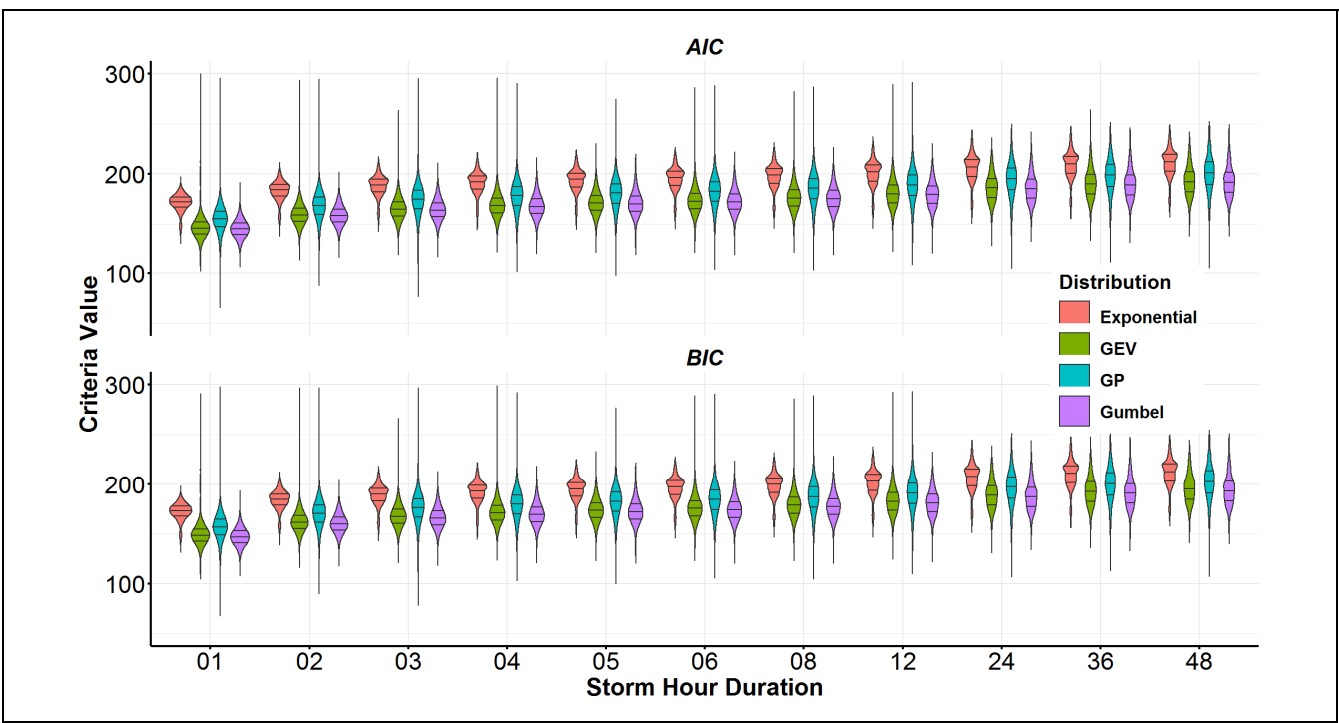

**Figure 5.** Violin plots showing the distribution of the AIC and BIC for each storm hour duration and each model (the three lines inside the violin represent the 25%, 50%, and 75% quartiles).

The spatial distribution of the best statistical distribution that fits the AMS supports the above conclusion. The overwhelming majority of radar pixels (about 90% of the state) indicate the best fit of GEV and Gumbel for all storm durations (Figure 6). Around 10% of the state, primarily located in the driest western area, show the best fit of the GP distribution according to AIC and BIC values in all duration hours. On average these areas receive less than 200 mm precipitation annually and are affected by monsoon events. Moreover, in the areas with the highest precipitation amounts (around the Houston metropolitan area), GEV is the best fitting distribution for long-duration storms (Figure 6C,D). The majority (almost three-quarters) of the state shows a strong goodness of fit with the Gumbel distribution for all durations (Figure 6). Perica [24] also found that GEV distribution fit the precipitation in the state of Texas better than the other families of distribution.

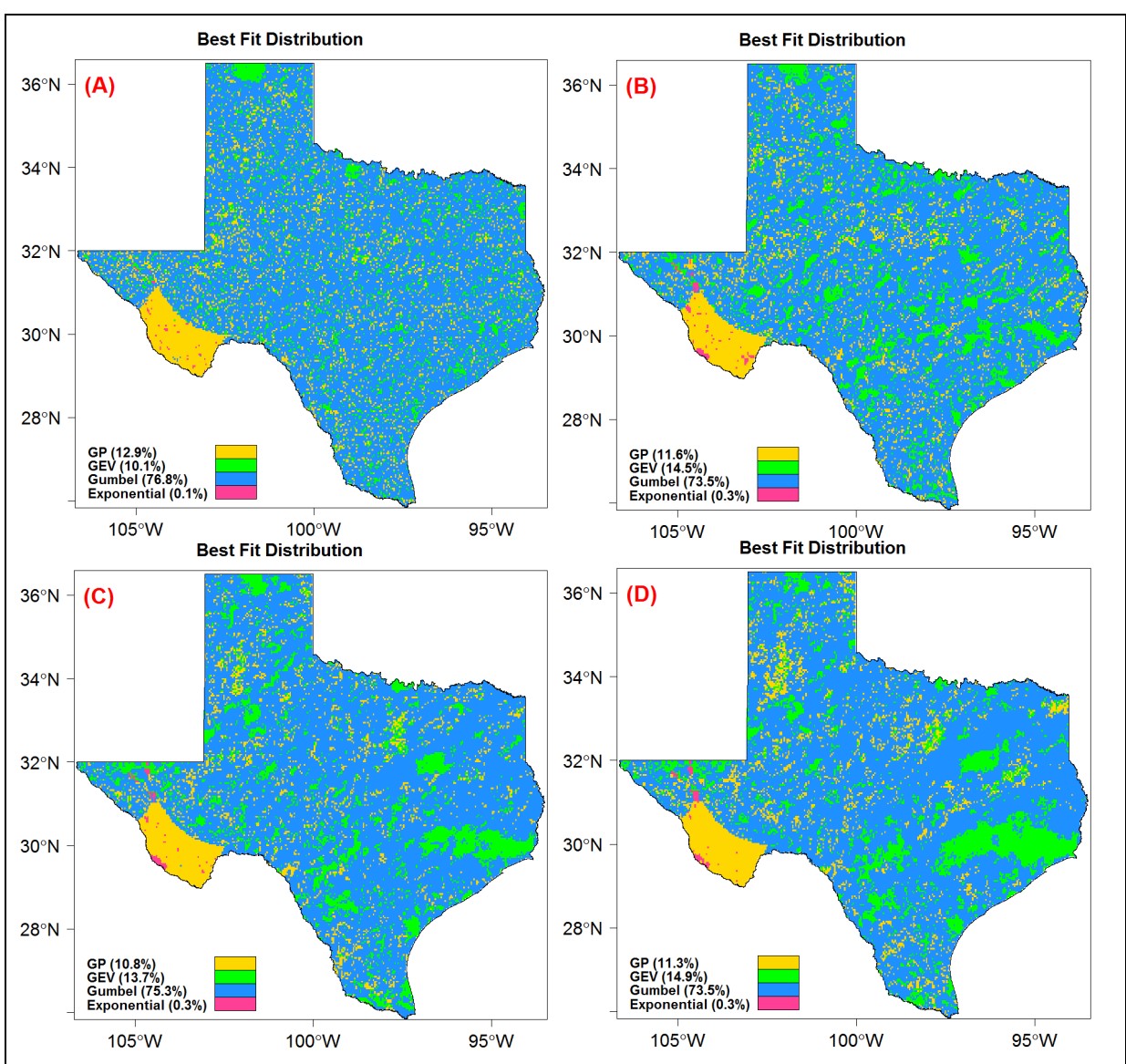

**Figure 6.** Spatial distribution of the best statistical model to fit the AMS for the (**A**) 1-h, (**B**) 6-h, (**C**) 24-h, and (**D**) 48-h storm durations.

### 4.3. Fitting the Gumbel Distribution

Based on the results of the AIC and BIC model selection criteria described in the previous section, the AMS was modeled using Gumbel distribution over the entire state. First, the location and scale parameters were estimated for all the 11 storm durations (Figures A1 and A2). The Kolmogorov–Smirnov test was used to test the fit of the Gumbel distribution. In more than 99% of the state, the fit was found to be significant at a 5% significance level for all storm durations (Figure A3). The Q-Q plots of the modeled and observed AMS quantiles show that the Gumbel distribution underestimates the lower quantiles and overestimates the higher quantiles in all of the metropolitan areas (Figure 7). Many validation studies that were conducted to assess the remotely sensed precipitation observed that the products underestimate heavy precipitation and overestimate light to medium precipitation [20,34–36], which is consistent with the results shown in Figure 7, with the exception of McAllen (Figure 7D). McAllen has the best fit and the lowest normalized root mean square error, while the worst fit was observed in Dallas-Fortworth, with a normalized root mean square error of 32%. The metropolitan areas that are located in a

closer proximity to the Gulf of Mexico show a better fit, with an NRMSE error of less than 15% (Figure 7B–D).

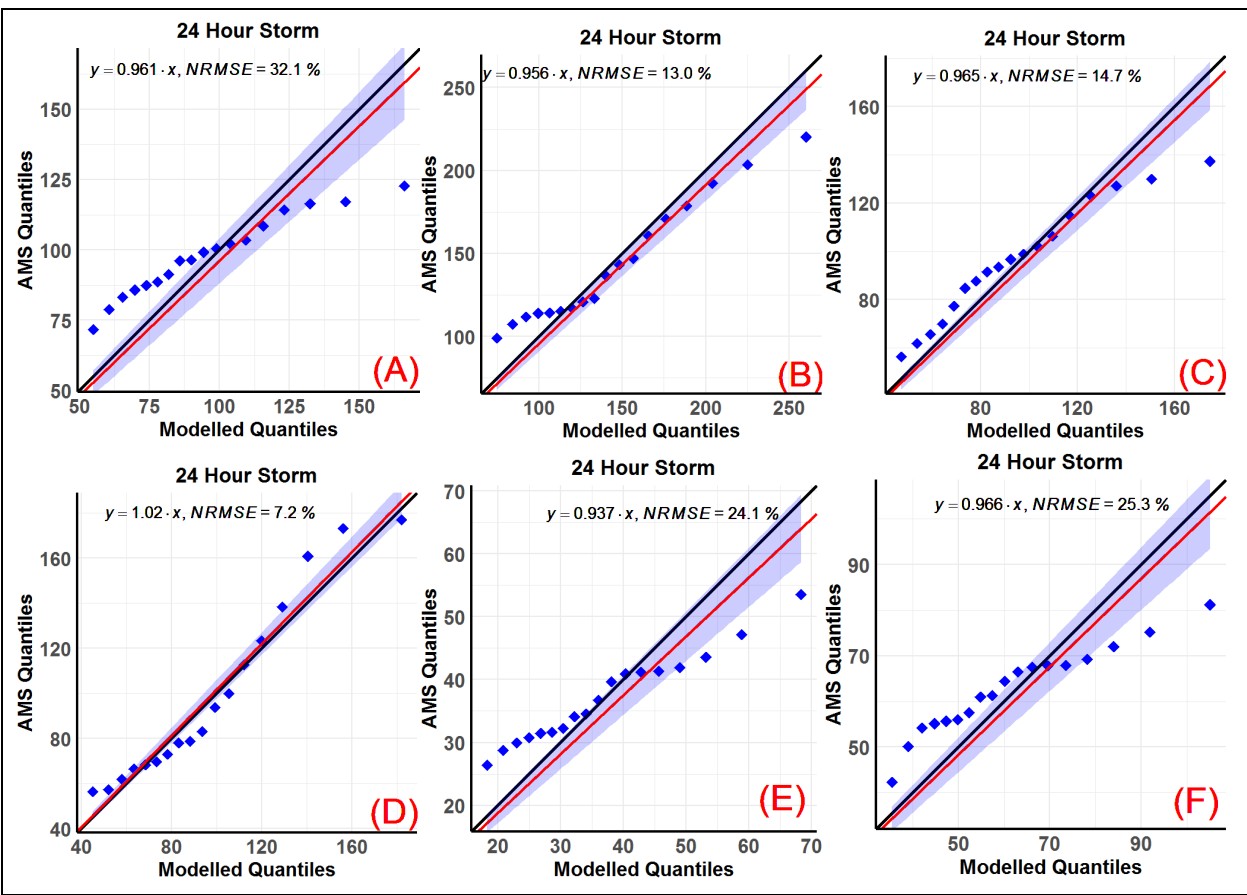

**Figure 7.** Q-Q plots of the AMS quantiles and Gumbel distribution with the regression line (red) and its 95% confidence computed using t-distribution for the 24-h storm duration for (**A**) Dallas-Fortworth, (**B**) Houston, (**C**) San Antonio, (**D**) McAllen, (**E**) El Paso, and (**F**) Amarillo (the quantiles used are 10%, 15%, 20%, . . . , 95%).

### 4.4. Development of the IDF Curves

The IDF curves were developed using the exceedance probability for radar pixels over the entire state. The IDF curves for all the six metropolitan areas are shown in Figure 8. Although only 19 years of data were available, the statistical model (Gumble) allows the extending of the estimates to higher return periods. The Houston metropolitan area, being the closest to the Gulf of Mexico, has the highest intensity values for all of the return periods (2-year return of 135 mm and 100-year return of 340 mm rainfall). McAllen and San Antonio (Figure 8C,D) have similar IDF curves for all return periods, with McAllen having slightly higher intensities. Surprisingly, Dallas-Fortworth precipitation maxima were found to be lower than those in McAllen and San Antonio, although Dallas average annual precipitation in the last two decades was estimated to be higher by approximately 200 mm. This shows that the extreme precipitation variability is highly influenced by the proximity to the Gulf of Mexico, which is a source of tropical cyclones. The lowest precipitation intensities were estimated in El Paso, with 34 mm for a 2-year return period and 90 mm for a 100-year return period, both for a 24-h storm event (Figure 8E). The spatial distribution of the precipitation frequencies for 2-year, 5-year, 10-year, 25-year, 50-year, and 100-year return periods are provided in Figures A4–A9, respectively.

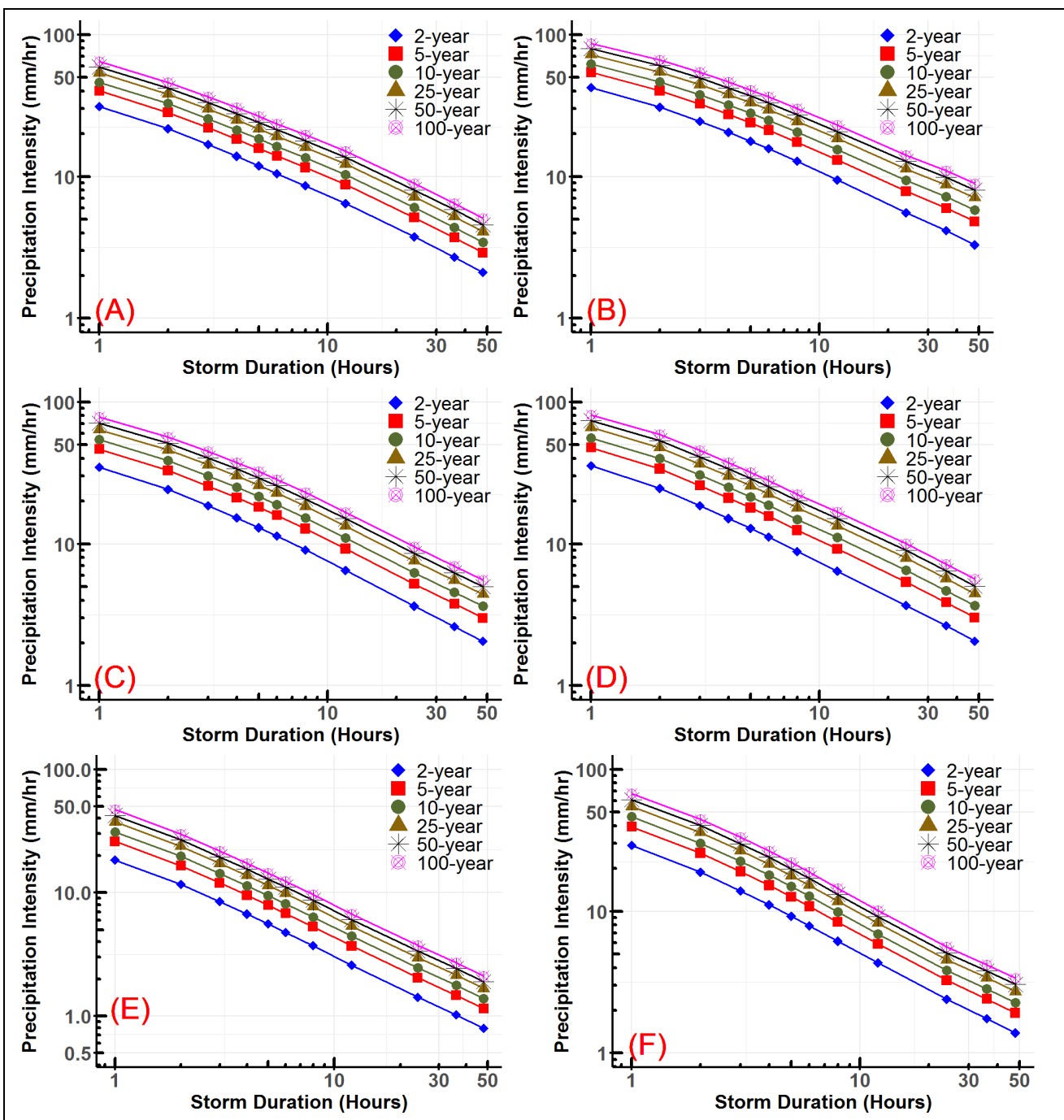

**Figure 8.** IDF curves for (**A**) Dallas-Fortworth, (**B**) Houston, (**C**) San Antonio, (**D**)McAllen, (**E**) El Paso, and (**F**) Amarillo (both *x*-axis and *y*-axis are in logarithmic scale for legibility of the curves).

*4.5. Comparison with the IDF Curves Developed by NOAA*

An evaluation of the NEXRAD Stage-IV-based IDF maps was conducted in relation to IDF maps from the NOAA Atlas 14 developed in 2018 [24] for the state of Texas. Compared to ATLAS 14 values, the NEXRAD Stage-IV-based IDF values show a difference that ranges within ±27% in over 95% of the state for all storm durations and return periods (Figure 9). An overwhelming portion of the state shows an underestimation of the 1-h duration storms, the smallest duration available with the NEXRAD Stage-IV product. This suggests that the Stage-IV radar data significantly underestimates high-intensity small-duration storm events more than those storm events with a longer duration due to the temporal averaging

of radar precipitation measurements at the top of the hour. Moreover, the Stage-IV-based IDF tends to underestimate values more for higher return period storms than for storms with smaller return periods in all the storm durations (Figure 9). For the 24-h storm duration, the precipitation intensity estimates show a difference within ±20% in over three-quarters of the state for all the return periods. The differences shown in Figure 9 are expected since NOAA estimates are based on gauge data at much lower spatial resolutions yet significantly higher temporal resolutions (15-min).

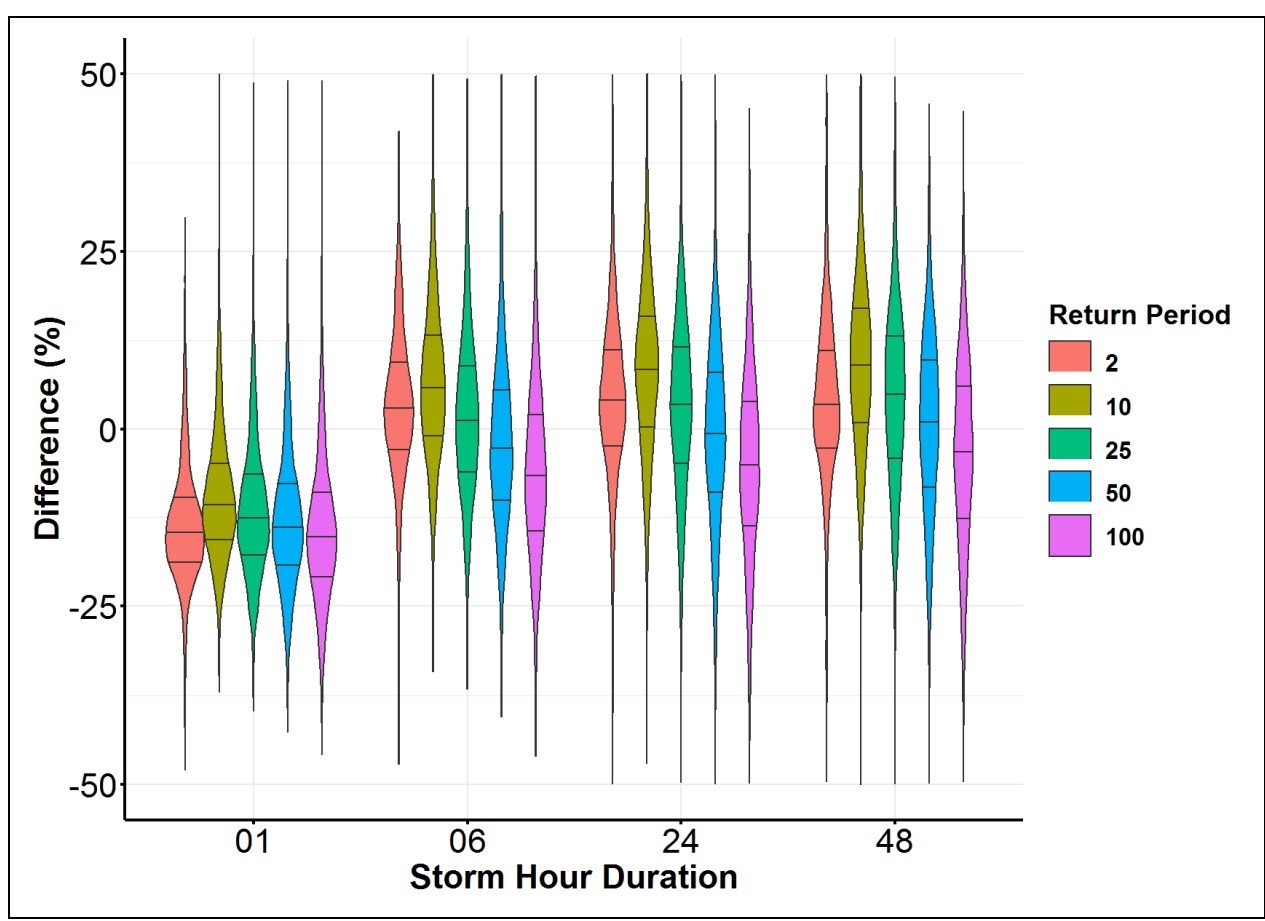

**Figure 9.** The distribution of the difference for all the storm durations and return periods across the state of Texas (Difference = (NEXRAD IDF value − NOAA Atlas 14 value)/NOAA Atlas 14 value).

The spatial distribution of the differences shows an overestimation by radar over the majority of the panhandle (north) and big bend (west) regions of the state and an underestimation over the southcentral areas for all storm durations (Figure 10). A comparison of the values for the 1-h duration 100-year return period storms shows underestimation over 90% of the state. Overestimation can only be seen in the northern and western regions (Figure 10A). Areas with a difference within ±10% for the 1-h storm duration for the 100-year return period cover about a quarter of the state. However, the area within a ±10% difference covers more than 50% of the state in the case of the 6-h, 24-h, and 48-h storm durations for the same return period. The average differences for the 24-h and 48-h storm durations are very similar, within ±1% for all return periods. This suggests that the IDFs for longer storm durations are almost the same.

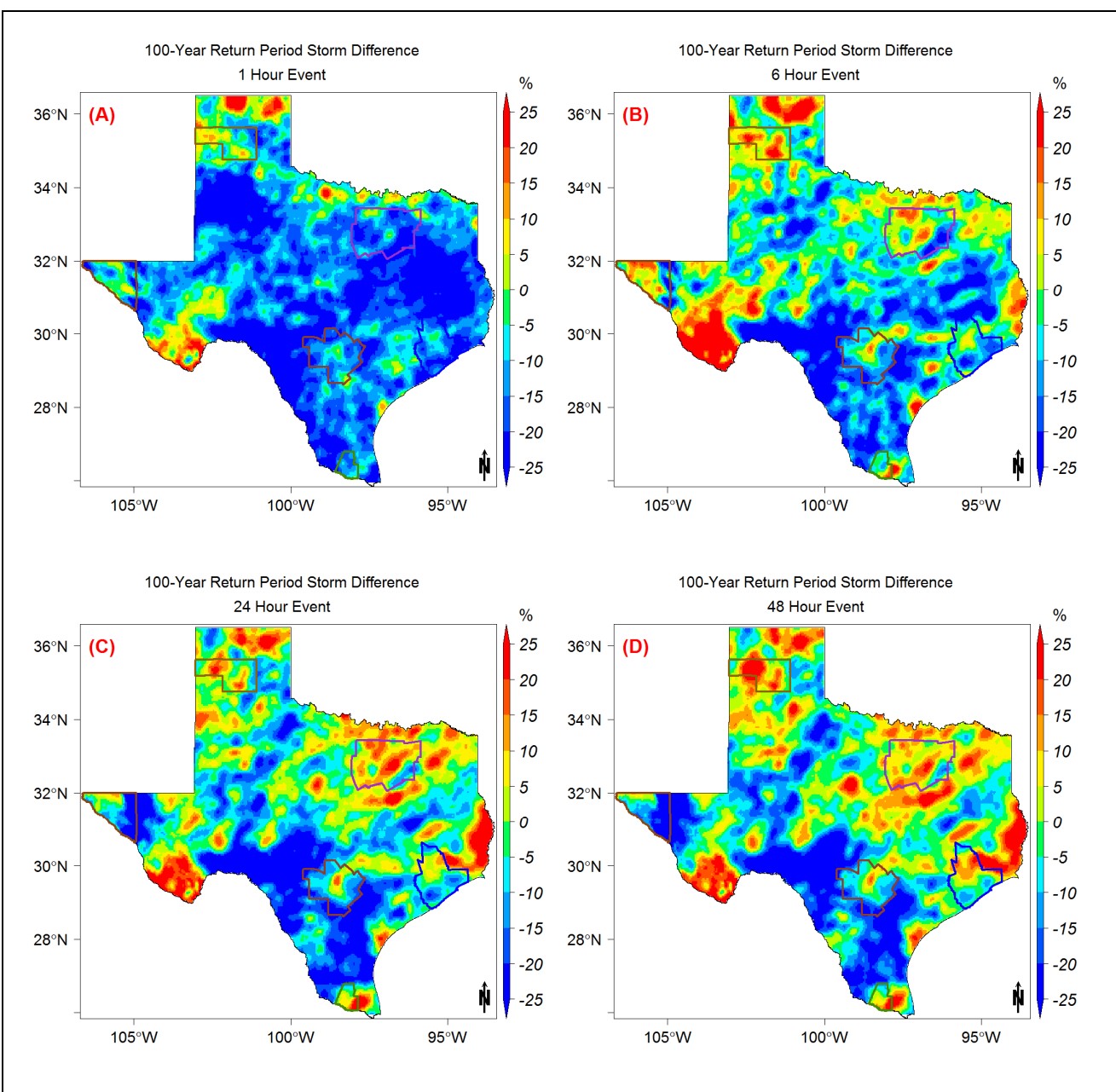

**Figure 10.** The distribution of the difference between NEXRAD and NOAA estimated intensities for the 100-year return period across the state of Texas for (**A**) 1-h, (**B**) 6-h, (**C**) 24-h, and (**D**) 48-h storm durations (Difference = (NEXRAD value − NOAA Atlas 14 value)/NOAA Atlas 14 value).

## 5. Summary and Conclusions

The changing climate is posing a significant threat to societies and infrastructure by increasing the frequency and magnitude of extreme weather across the globe. The rise in the frequency of extreme flooding is caused by increased evaporation due to rising atmospheric temperature and the capacity of the warming atmosphere to hold moisture. This study developed IDF curves using NEXRAD Stage-IV precipitation data for the state of Texas at a high spatial resolution. The timeframe of the study covers 19 years ranging from 2002 to 2020. The NEXRAD data was used to estimate the Annual Maxima Series (AMS) for different storm durations. GEV, Gumbel, GP, and Exponential distributions were used to fit the AMS. The best model was selected using the Akaike Information Criterion (AIC) and the Bayesian Information Criterion (BIC) criteria. The Kolmogorov–Smirnov test

was used to test the significance of the distribution model. The null hypothesis ($H_o$) states that the two dataset values are from the same continuous distribution, while the alternative hypothesis ($H_a$) states that these two datasets are from different continuous distributions.

The Gumbel distribution was found to have the lowest AIC and BIC for over three-quarters of the state area for all storm durations. For around 10% of the state, primarily in the driest western part, GP was the best distribution for fit the AMS according to AIC and BIC for all storm durations. In more than 99% of the state, the Gumbel distribution fit was acceptable, with a significance level of 5% for all storm durations. The Gumbel distribution seems to be suitable for semi-arid areas according to the spatial distribution of best model fit. GP was found to be more convenient for arid regions. The modeled quantiles and AMS quantiles show that Gumbel distribution underestimates the lower quantiles and overestimates the higher quantiles in the major metropolitan areas of Texas. The northern coastal areas of the Gulf of Mexico have the highest precipitation intensities for all storm durations and return periods. Houston has the highest IDF values for all storm durations and return periods (2-year return of 135 mm and 100-year return of 340 mm rainfall). The lowest IDF values were estimated in the El Paso metropolitan area, with 34 mm for a 2-year return period and 90 mm for a 100-year return period, both for a 24-h storm event. Interestingly, the Dallas-Fortworth IDF precipitation intensities were found to be lower than those in McAllen and San Antonio although the average annual precipitation for Dallas in the last two decades was higher by approximately 200 mm. Evaluation of the developed IDF curves was conducted by comparison against the latest NOAA Atlas 14 values released in 2018. The radar-based IDF values showed differences within $-27\%$ to 27% in over 95% of the state for all storm durations and return periods. This clearly shows the potential of remotely sensed precipitation products in developing IDF curves, especially when longer records are available. Moreover, the Stage-IV-based IDF curves tend to particularly underestimate the intensities of higher return periods, which may be related to the precipitation threshold applied to rainfall estimation. In the 24-h storm duration, the precipitation frequency estimates show differences within $\pm 20\%$ in over three-quarters of the state for all the return periods.

The main limitation of this study is the length of the data, which is limited to only 19 years. Moreover, the discontinuity in the data is another problem with the Stage-IV data, as the datasets were developed separately by the 12 River Forecast Centers (RFC) and then mosaicked into a single product to cover the conterminous US; however, these discontinuities are very minor. The main strength of the analysis is the employment of a very high ($4 \times 4$ km) spatial resolution precipitation product. The developed IDF maps can be used in conjunction with Atlas 14 maps to have a better idea of the spatial variability within any area of the state and eliminate the need for area reduction factors.

**Author Contributions:** H.O.S. guided this research and contributed significantly to the preparation of the manuscript for publication (with input from D.T.G.). D.T.G. downloaded and processed the remote sensing products. D.T.G. developed the research methodology (with input from H.O.S.). D.T.G. developed the scripts used in the analysis and performed the statistical analysis. D.T.G. prepared the first draft. H.O.S. performed the final overall proofreading of the manuscript. Both authors have read and agreed to the published version of the manuscript.

**Funding:** This research received no external funding.

**Institutional Review Board Statement:** Not applicable.

**Informed Consent Statement:** Not applicable.

**Data Availability Statement:** Publicly available datasets were analyzed in this study. NEXRAD Stage-IV data presented in this study are openly available at [https://data.eol.ucar.edu/dataset/21.093] at [doi:10.5065/D6PG1QDD]. Assesse Date (20 March 2021).

**Conflicts of Interest:** The authors declare no conflict of interest.

## Appendix A

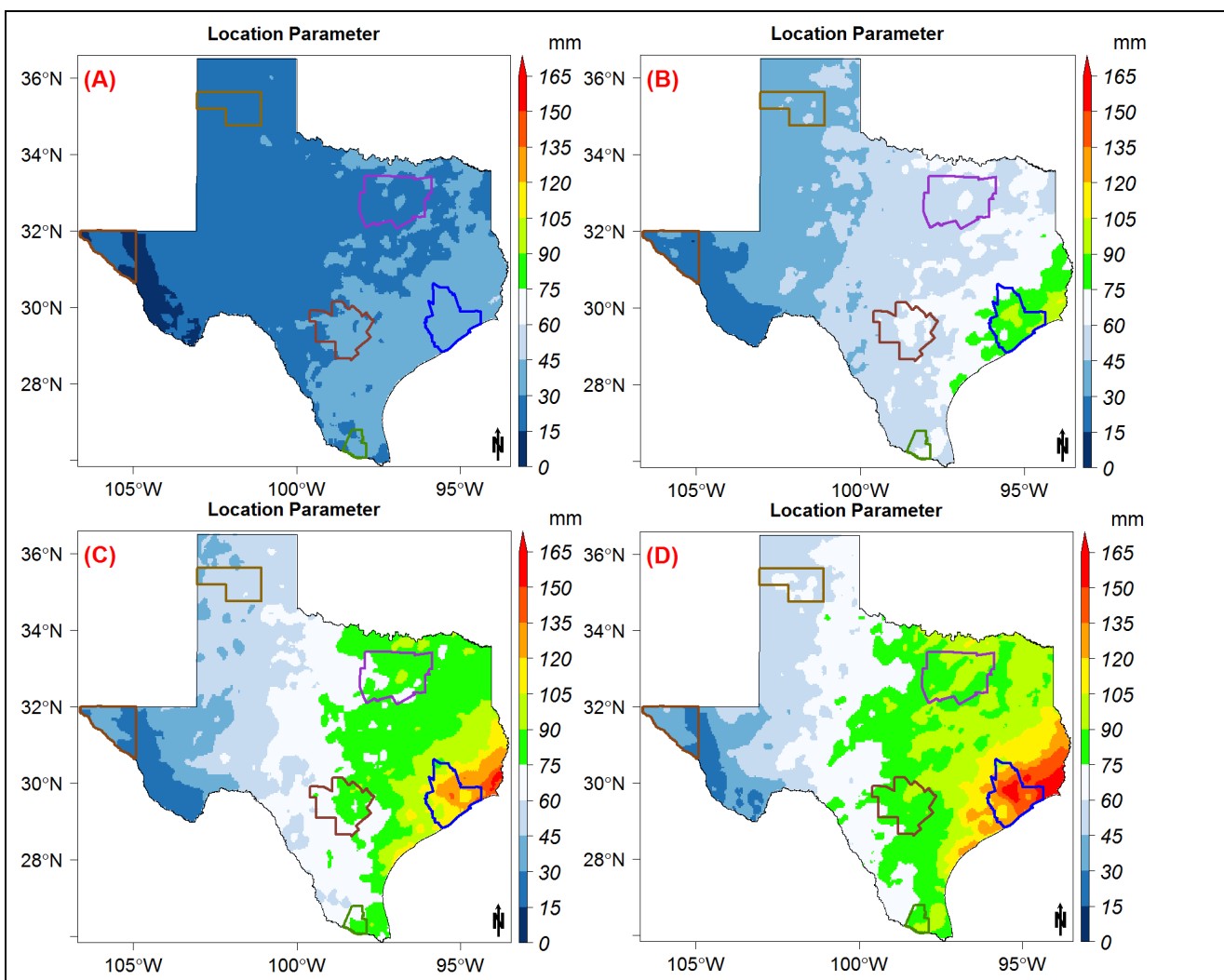

**Figure A1.** Location parameters of the Gumbel model distribution for the (**A**) 1-h, (**B**) 6-h, (**C**) 24-h, and (**D**) 48-h storm durations.

## Appendix B

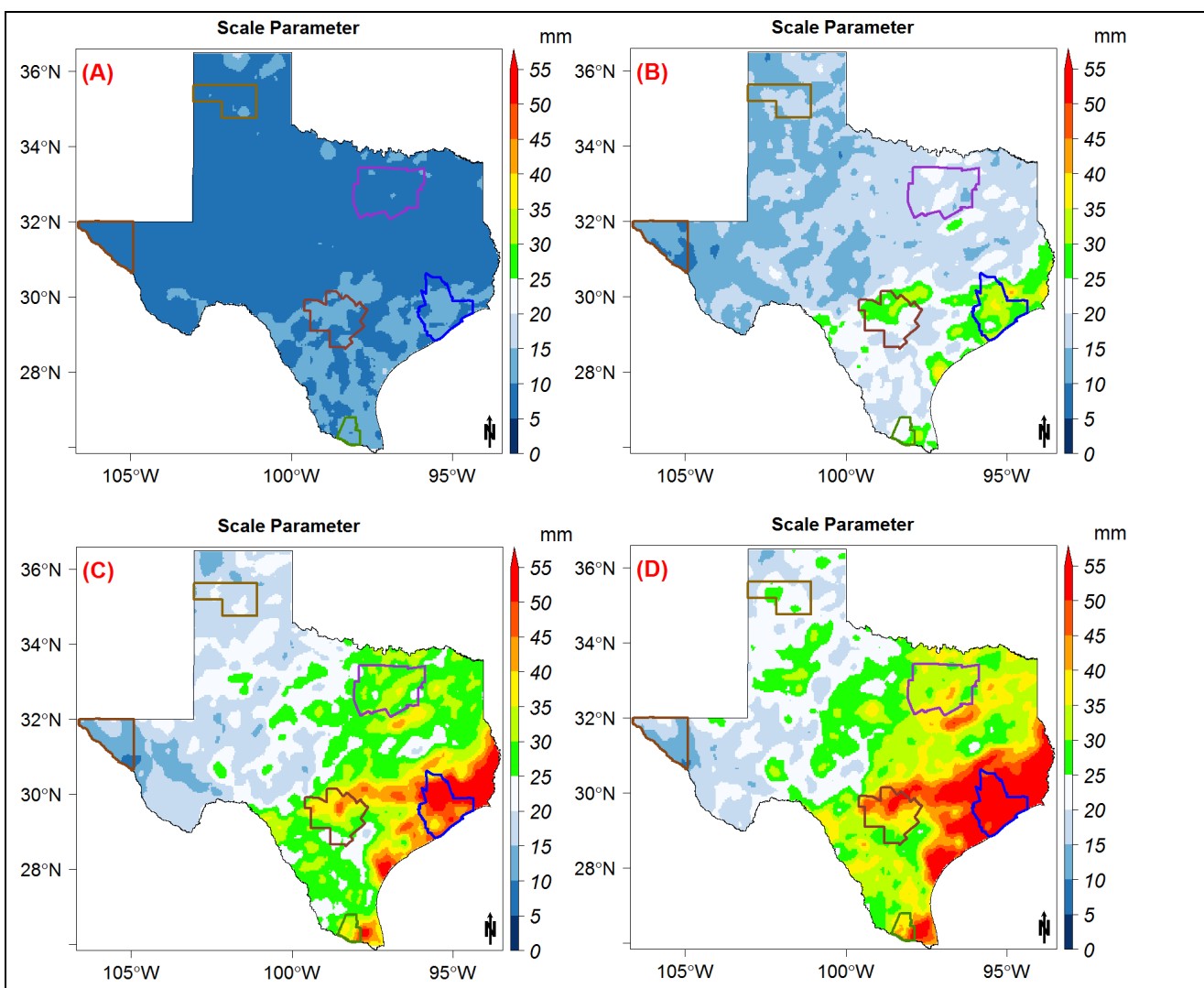

**Figure A2.** Scale parameters of the Gumbel model distribution for the (**A**) 1-h, (**B**) 6-h, (**C**) 24-h, and (**D**) 48-h storm durations.

## Appendix C

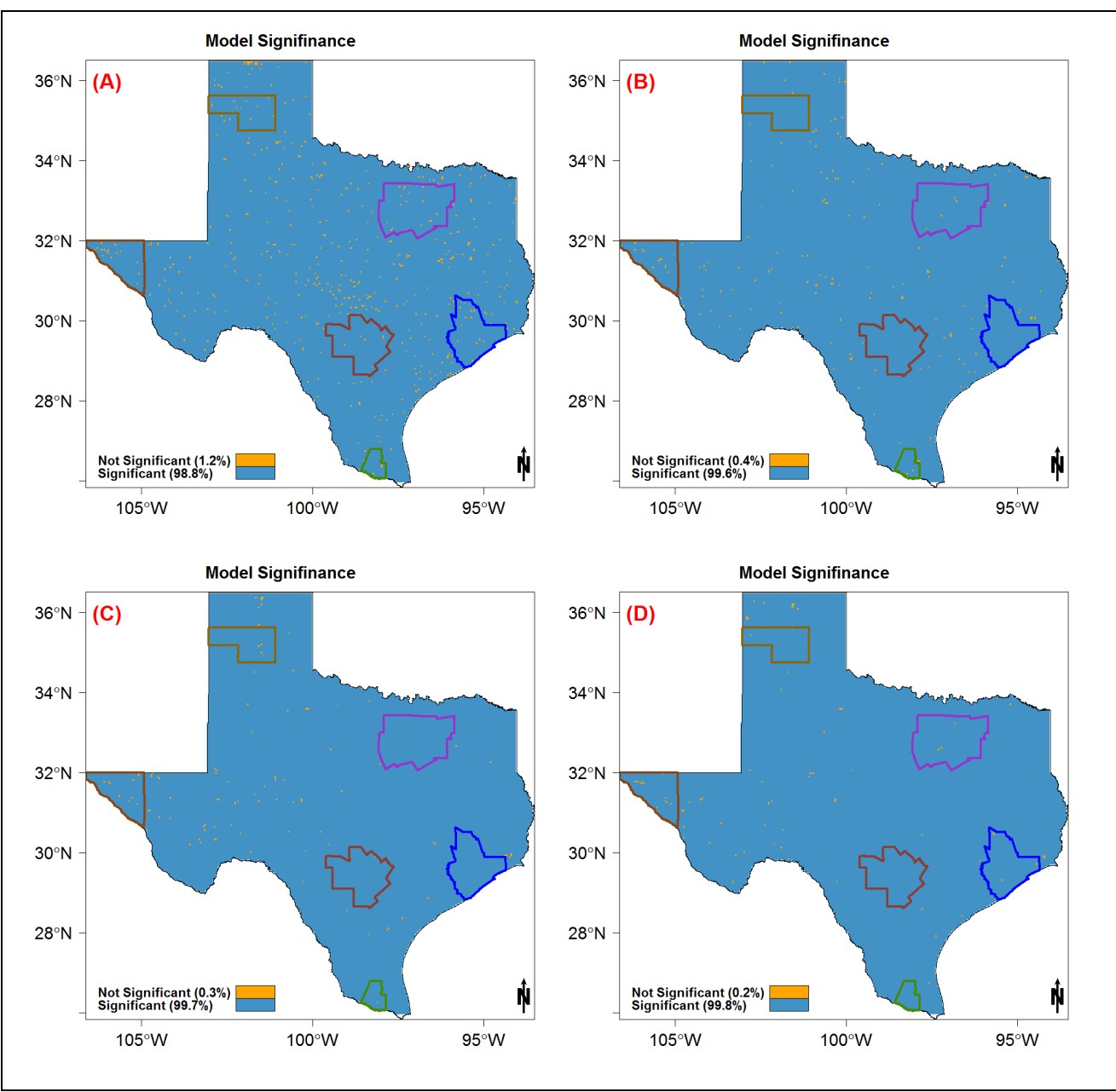

**Figure A3.** Spatial distribution of the Kolmogorov–Smirnov model of significance test for the Gumbel model distribution for the (**A**) 1-h, (**B**) 6-h, (**C**) 24-h, and (**D**) 48-h storm durations.

## Appendix D

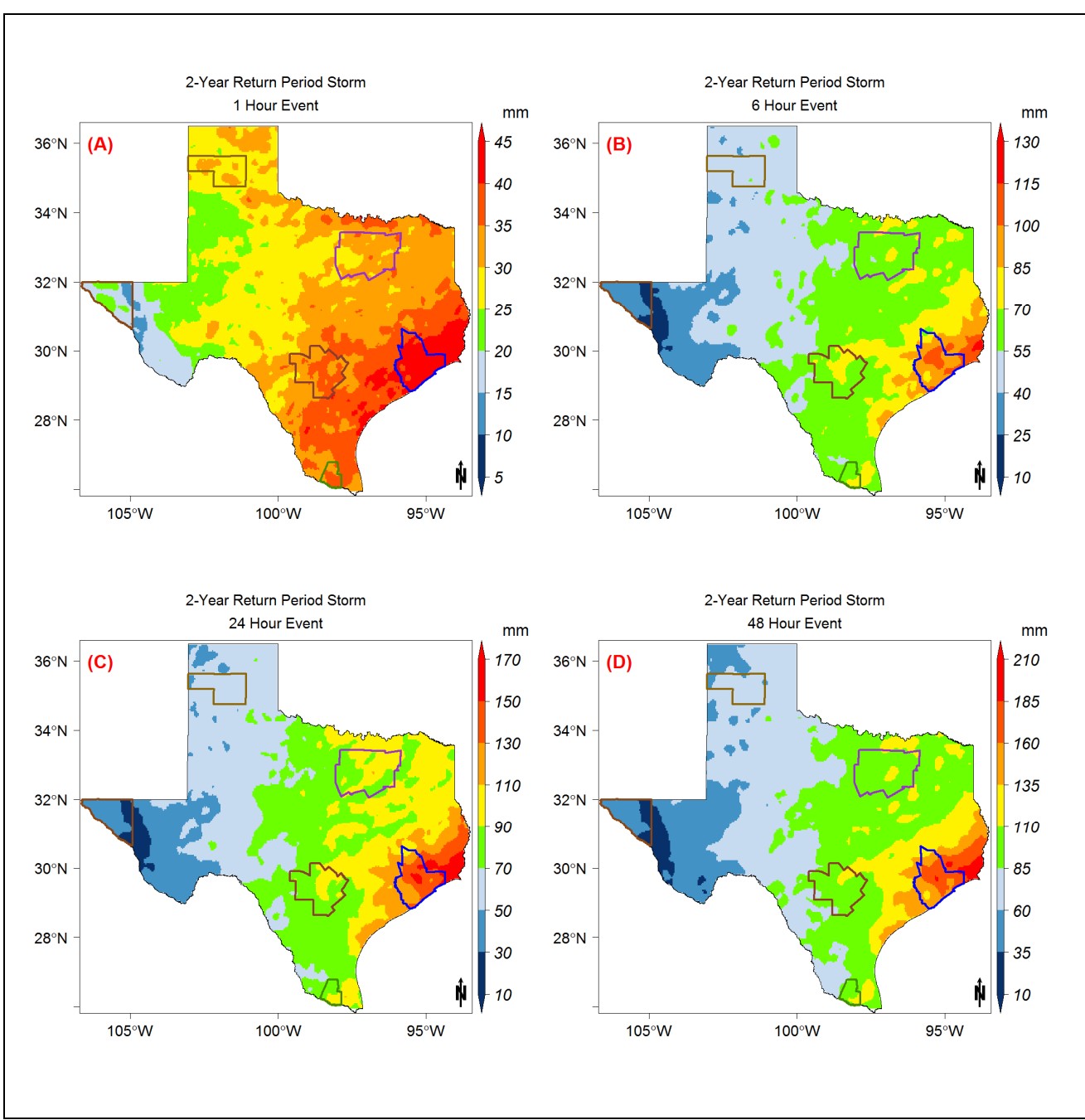

**Figure A4.** The 2-year return period storm event map modeled using the Gumbel model distribution for the (**A**) 1-h, (**B**) 6-h, (**C**) 24-h, and (**D**) 48-h storm durations.

## Appendix E

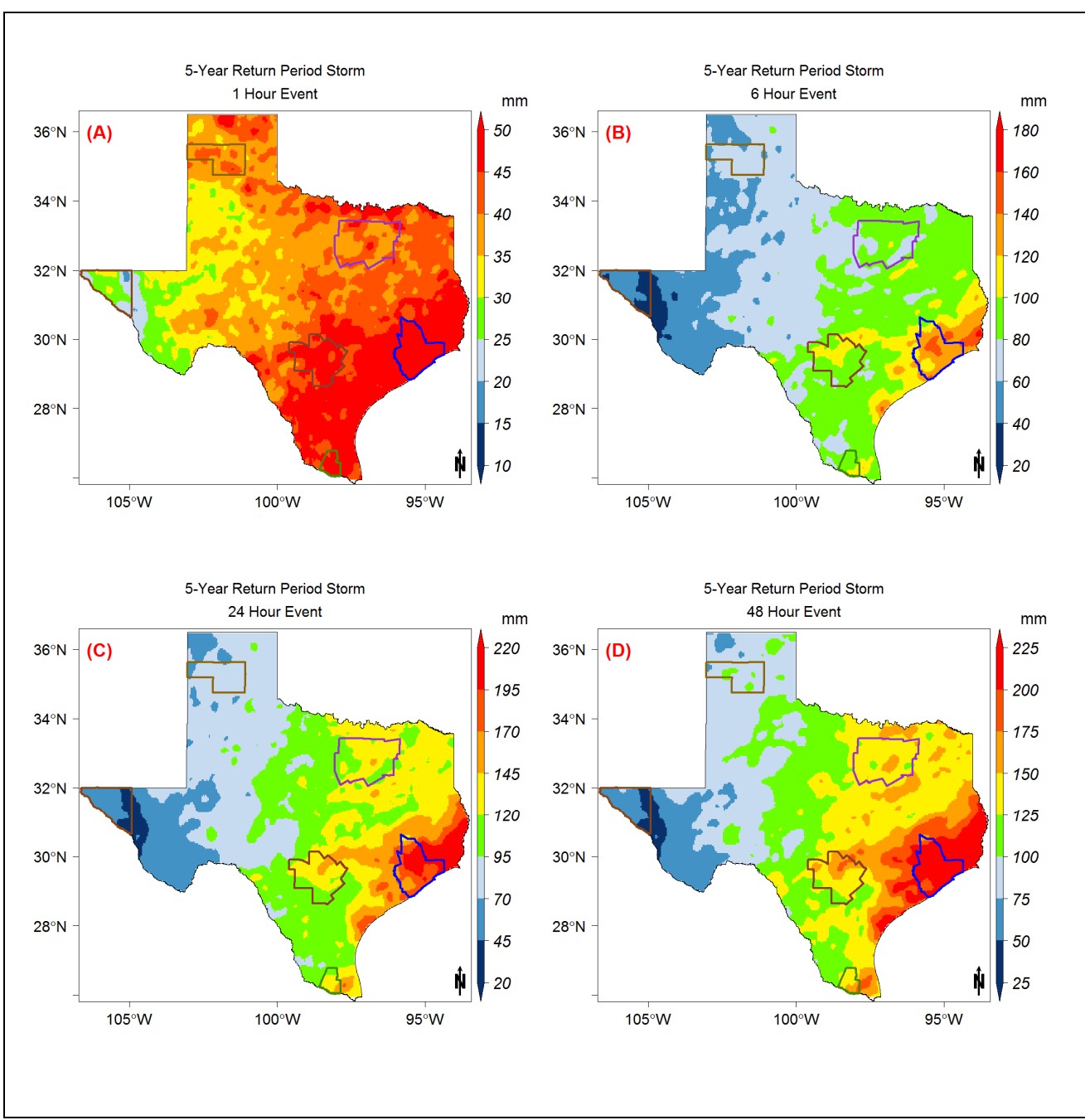

**Figure A5.** The 5-year return period storm event map modeled using the Gumbel model distribution for the (**A**) 1-h, (**B**) 6-h, (**C**) 24-h, and (**D**) 48-h storm durations.

## Appendix F

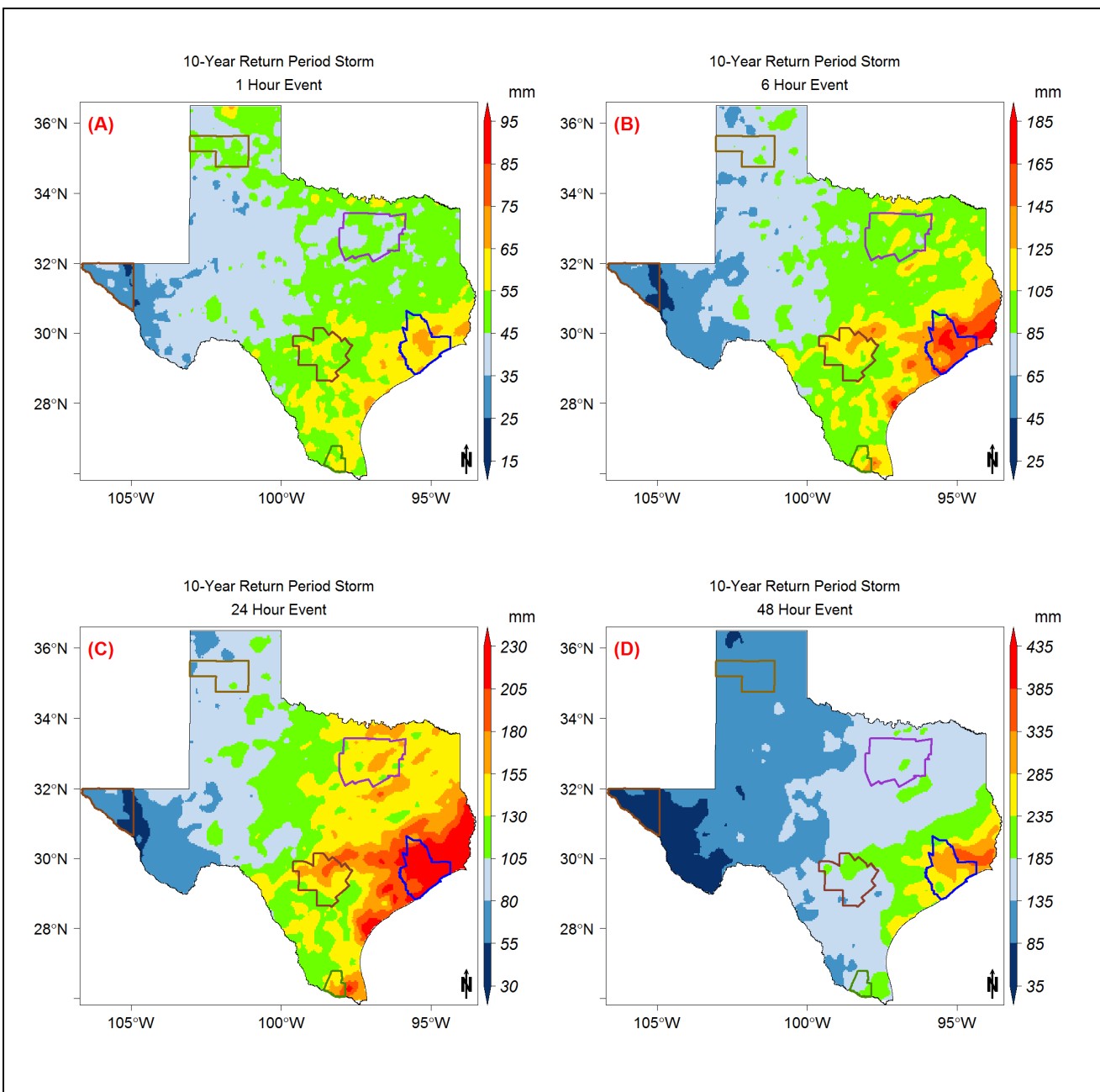

**Figure A6.** The 10-year return period storm event map modeled using the Gumbel model distribution for the (**A**) 1-h, (**B**) 6-h, (**C**) 24-h, and (**D**) 48-h storm durations.

## Appendix G

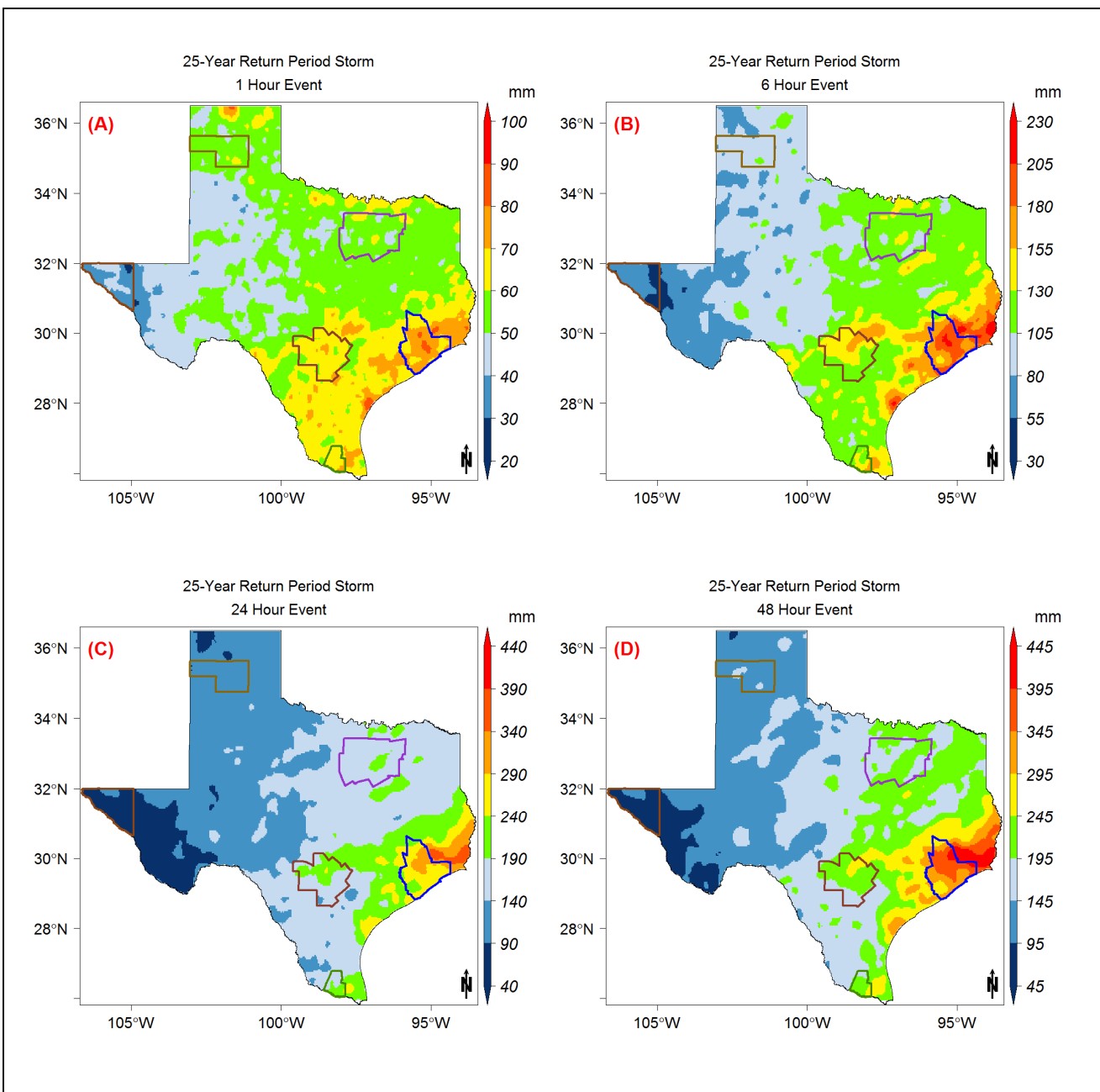

**Figure A7.** The 25-year return period storm event map modeled using the Gumbel model distribution for the (**A**) 01-h, (**B**) 6-h, (**C**) 24-h, and (**D**) 48-h storm durations.

## Appendix H

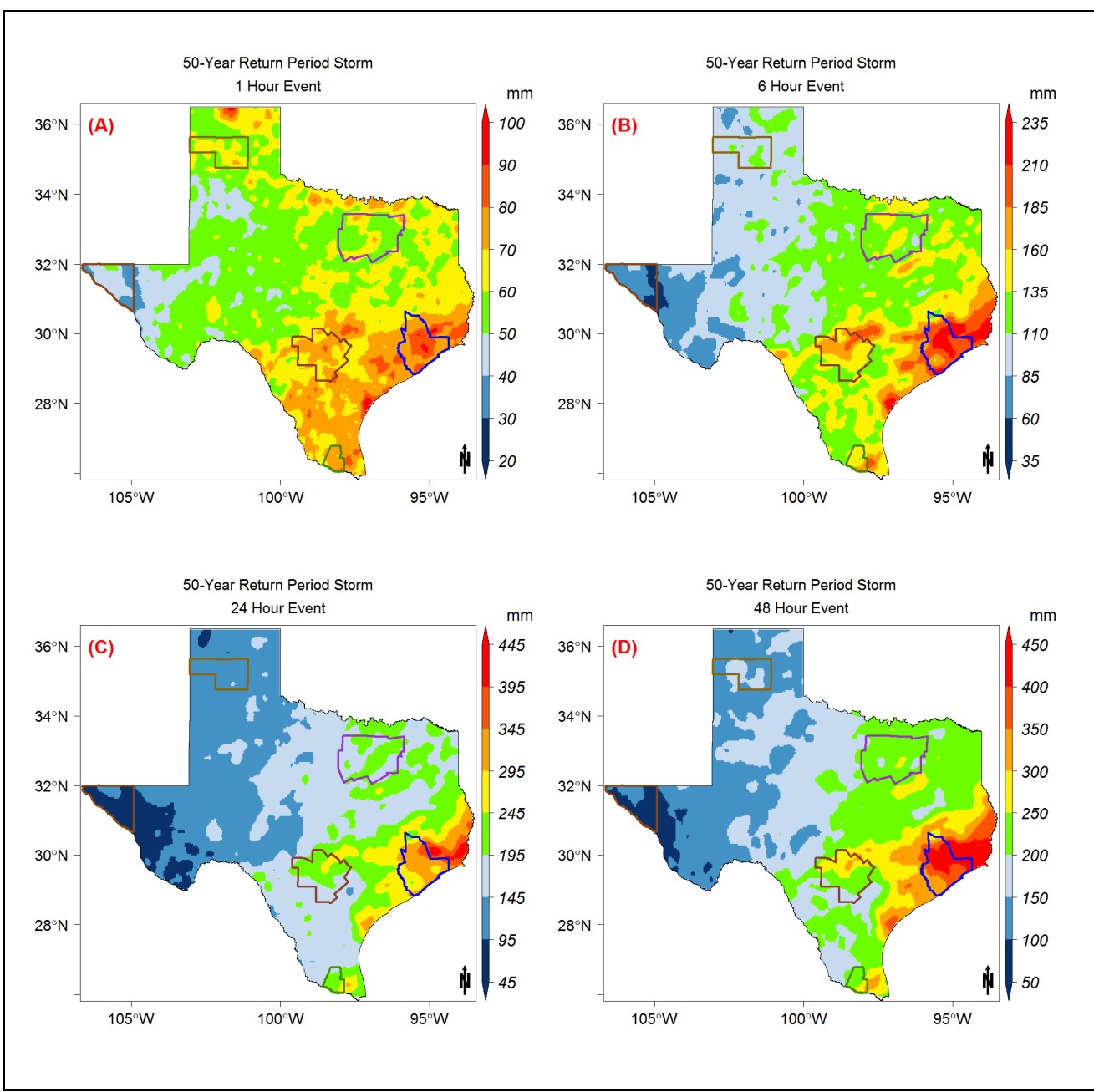

**Figure A8.** The 50-year return period storm event map modeled using the Gumbel model distribution for the (**A**) 1-h, (**B**) 6-h, (**C**) 24-h, and (**D**) 48-h storm durations.

## Appendix I

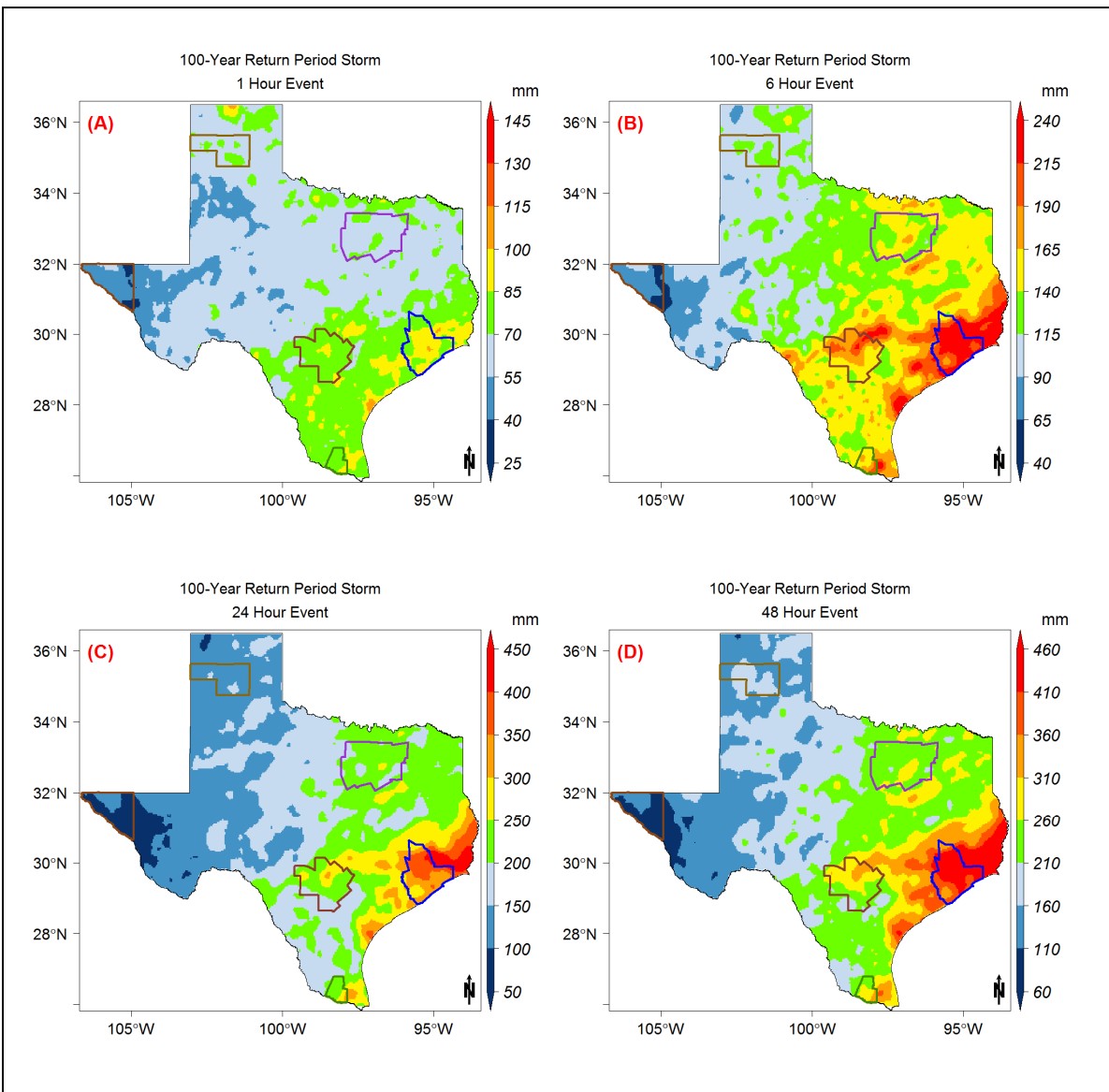

**Figure A9.** The 100-year return period storm event map modeled using the Gumbel model distribution for the (**A**) 1-h, (**B**) 6-h, (**C**) 24-h, and (**D**) 48-h storm durations.

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
