# Peer review of "Development and Assessment of High-Resolution Radar-Based Precipitation Intensity-Duration-Curve (IDF) Curves for the State of Texas"

_remotesensing, doi:10.3390/rs13152890_

Round 1
Reviewer 1 Report
June 30, 2021
Manuscript: ‘Development and Assessment of High-resolution Radar-based Precipitation Intensity-Duration-Curve (IDF) Curves for the state of Texas’
In this paper, IDF curves were derived from the NEXRAD Stage-IV high-resolution radar data over the entire state of Texas for the period 2002-2020. Unlike other similar works, the NEXRAD-based annual Maximum Series (AMS) were fitted to four theoretical extreme value statistical distributions, and then validated against the operational Atlas 14 IDF. This is a relevant topic lies within the scope of the MDPI remote sensing journal. The article is very well organized and neatly written with the appropriate scientific content. Based on the above, I support the publication of this manuscript, but only after a minor revision.
********************************
Title: it fits perfectly the paper content.
Abstract: it is quite adjusted to the paper content.
Lines 23: I would like to suggest that authors add a numeric metric to support these findings. For example, the mean difference per region indicated in lines 356-367.
Introduction: it provides sufficient background and includes relevant references on different statistical approaches to obtain IDF curves, highlighting their main limitations and strengths and the need to evaluate high-resolution spatio-temporal precipitation products in the state of Texas. Objectives and the novelty are clearly presented.
Study Area and Dataset: the study area, datasets are clearly described.
Methodology: this section has been clearly described.
Results and Discussion: these are clearly presented.
Line 246: in this point, I would suggest that authors apply the Mann-Kendall trend test to assess the significance of trends for 24-hour AMP time series in Dallas-Fortworth, Houston, San Antonio, McAllen, El Paso, and Amarillo.
Line 292, Figure 6: for clarity, I would suggest that authors use different colors to improve the visual discrimination between statistical models
Summary and Conclusions: this is clearly presented and is supported by results from the previous section.
Line 405: fix ‘…longer records are available’
Author Response
Dear Reviewer,
Please see the attachment for our response.
Best Regards,
Manuscript Authors

Reviewer 2 Report
Overview
Some clarifications need in order to have a full comprehension of the proposed study, in particular about the comparison between the IDF curves based on the remote sensing source (i.e., NEXRAD) and raingauges (i.e., NOAA ATLAS).
The major limitation of the present study (as authors acknowledged in the conclusions) is the limited lenght of NEXRAD dataset used to estimate IDF curves.
Broad comments
In the introduction, authors highlight limitations that generally affect IDF curves based on raingauges (for instance, at lines 53-58, 99-101) and datasets based on remote sensing sources (for instance, at lines 82-86). By reading the contents of the introduction, a reader could interpret that the research described in the present manuscript allows to overcome these issues. However, many of these limitations still characterize outcomes of the present paper. The general sense of the introduction should be refined.
Specific comments
Lines 53-58: the three mentioned limitations (i.e., sparseness of the rain gauge network, historical records often differs significantly from one rain gauge to another, spatial variability of IDF) should be also discussed with respect to the IDF curves provided by NOAA.
Lines 82-84: this limitation is valid also for the dataset available for the present study.
Lines 84-86: this issue is valid also for the dataset available for the present study.
Lines 99-101: results for not densely populated area should be evidenced in order to overcome the uneven distribution of rainguages; but, results for cities (i.e., densely populated areas) are highlighted in this study.
Lines 112-114: was the comparison performed over the same period? Were IDF curves from NOAA computed on the same data period used for IDF curves from NEXRAD?
Lines 201-203: some clarifications should be added to make more clear the described process. Which set of parameters was used to select the suitable distribution with the best goodness? How is the model used in the evaluation of the best fit if the optimized parameters are estimated after the selection of the best fit?
Lines 206-207: does the cited pixel in the moving window refer to the 4x4 km nexrad grid?
Line 209: make explicit the acronym AMS. It was defined just in the abstract.
Lines 213-215: specify over which grid the comparison was performed. On the 4x4 nexrad grid or over the more than 2000 points of NOAA IDF curves? I mean, the noaa idf curves were re-scaled over the nexrad grid? Or the nexrad grid point nearest to the ATLAS rainguage was used?
Lines 218-219: specify if the comparison is performed with different dataset periods for the same grid point.
Line 227: check if an additional blank space is added after “Figure 3.”
Lines 242-243: specify how many grid points were used for each metropolitan area and cite Figure 4 in the text.
Lines 258-259: specify how the 95% confidence interval is computed.
Line 265: check if an additional blank space is added after “Figure 5.”
Lines 265-267: specify how the distribution is obtained (by considering all the pixels of the nexrad grid?).
Line 270: where (or how) is the mean of the distribution displayed in Figure 5?
Lines 274-276: it is not clear how this statement can be supported by equations of AIC and BIC. Please clarify the link between the increase of AIC and BIC and precipitation amount.
Line 301: the AMS quantiles should be identified.
Lines 300-302: this outcome is not valid for panel D.
Lines 302-305: this statement should be also supported by an analysis performed with the dataset used in the present study (i.e., nexrad versus raingauges).
Lines 311-312: specify how the 95% confidence interval is computed.
Line 332: in panel E, it seems that the point of the higher duration for the 2-year return period is not displayed.
Lines 335-338: specify over which grid the comparison was performed. On the 4x4 nexrad grid or over the more than 2000 points of NOAA IDF curves?
Lines 340-343: it is not clear how this statement can be supported by the mentioned reason. Please clarify why the temporal averaging at the top of the hour has a different impact for short and long durations.
Lines 347-349: it is not clear the reason for which the NWS HYDRO-35 report is cited. The citation appears as off-topic.
Line 373: check if an additional blank space is added after “the globe.”
Line 404: check if an additional blank space is added after “curves”
Lines 406-407: this statement is not clear. To which precipitation thresholds are authors referring to?
Line 414: check if an additional blank space is added after “minor.”
Lines 415-417: this statement could be questionable, given that details lack throughout the manuscript about how the comparison between nexrad- and noaa-based IDF curves was performed (same data period? verification points or grid? Please refer to specific comments for lines 112-114, 213-215, 218-219, 335-338). In addition, the issue about the need or not of the area reduction factor was not discussed.
Author Response
Dear Reviewer,
I would like to appreciate your detailed feedback about our work. Personally, this is the best review that I have received so far in my short career. I would like to appreciate your input and your valuable time.
Please see the attachment for our response.
Best Regards,
Dawit Ghebreyesus

Reviewer 3 Report
The paper presents IDF curves developed with the help of NEXRAD stage-IV precipitation data for the state of Texas. It is a suitable topic for Remote Sensing MDPI journal. Therefore, I would recommend a revision following my comments below.
GENERAL COMMENTS:
- The methodology should be more explored/developed.
- It could be (better) discussed in the Summary and Conclusions how this study could be applied elsewhere (and not only in Texas) and whether the selected distribution (in this case, Gumbel distribution) could be also applied elsewhere based on this study (or if the same analysis should be performed for other areas to select the better distribution).
SPECIFIC COMMENTS:
- Lines 67-74: Do the authors consider 10km-30min and 8km-30min “high” spatio-temporal resolutions? Compared to the recent spatio-temporal resolutions achieved by ground-based weather radars, the cited resolutions would not be considered as “high” resolutions. Please revise it.
- Section 2.2: A figure of the NEXRAD mosaic could be presented in this section, if possible, with the weather radars’ locations (especially those covering Texas state area).
- Lines 189 and 199: no indentation is needed.
- Figure 2: Wouldn’t the period of hourly data be 2002-2020 (as mentioned at lines 158-159, 226-227, and 377-378) instead of 2002-2019 (as presented in the figure)? Please revise it.
- Lines 232-239: The 12-hour storm duration is broadly mentioned and discussed in these sentences; however it is not present in Figures 3, 6, 9, 10, and all appendix figures. I suggest the authors to include this storm duration in the mentioned Figures and associated analyses.
- Figure 4 was not mentioned in the text.
- Figures from Appendix should be numbered accordingly (e.g., Figure A.1, Figure B.1, …) and mentioned in the text.
Author Response
Dear Reviewer,
Please see the attachment for our response.
Best Regards,
Dawit Ghebreyesus

Round 2
Reviewer 3 Report
The paper presents IDF curves developed with the help of NEXRAD stage-IV precipitation data for the state of Texas. It is a suitable topic for Remote Sensing MDPI journal. The authors have performed a satisfactory review of the manuscript. However, I would still recommend a minor revision following my only comment below.
- The methodology has not been sufficiently explored/developed.
Author Response
Dear Reviewer,
I would like to thank you again for your constructive comments and for the time you spent to enhance our work.
Please find our response in the attachment.
Regards,
Authors
